# METRIC-NORMALIZED POSTERIOR LEAKAGE (MPL): ATTACKER-ALIGNED PRIVACY FOR JOINT CONSUMPTION

## ABSTRACT

*Metric differential privacy (mDP)* strengthens *local differential privacy (LDP)* by scaling noise to semantic distance, but many ML systems are consumed under *joint observation*, where model-agnostic, per-record guarantees can miss leakage from evidence aggregation. We introduce *metric-normalized posterior leakage (mPL)*—an attacker-aligned, distance-calibrated measure of posterior-odds shift induced by releases—and show that for single or independent releases, uniformly bounding mPL is equivalent to mDP. Under joint observation, however, satisfying mDP may still leave mPL high because learned aggregators compound evidence across correlated items. To make control practical, we formalize *probabilistically bounded mPL (PBmPL)*, which limits how often mPL may exceed a target budget, and we operationalize it via *Adaptive mPL (AmPL)*, a *trust-and-verify* pipeline that perturbs, audits with a learned attacker, and adapts parameters (with optional Bayesian remapping) to balance privacy and utility. In a word-embedding case study, neural adversaries violate mPL under joint consumption despite per-record mDP perturbations, whereas AmPL substantially lowers the frequency of such violations with low utility loss, indicating PBmPL as a practical, certifiable protection for joint-consumption settings.

## 1 INTRODUCTION

*Local Differential Privacy (LDP)* Duchi et al. (2013) enforces a uniform indistinguishability notion over any two inputs. That metric-agnostic view clashes with modern continuous and embedding spaces Imola et al. (2022): distances encode semantics (nearby points are similar; far ones are not). Treating all pairs equally either injects excess noise (hurting downstream utility) or under-protects fine-grained neighborhoods.

Metric-aware privacy incorporates geometry into the guarantee. Early formulations, such as *metric DP (mDP)* Chatzikokolakis et al. (2013) (also known as *Lipschitz privacy* Koufogiannis et al. (2015)), require that closer secrets remain harder to distinguish than distant ones, which proved natural for locations and embeddings. In location privacy, *geo-indistinguishability* Andrés et al. (2013) popularized practical mechanisms (e.g., planar Laplace noise), followed by optimization-based obfuscation tailored to road networks, points of interest, and mobility priors. Subsequent work Bordenabe et al. (2014); Liu & Qiu (2025) studied optimal mechanism design under metric constraints, linked utility to transport/Wasserstein costs, and adapted composition/group privacy to metric spaces. Beyond coordinates, mDP ideas extend to representation spaces (text/image embeddings) and to learning pipelines that consume batches or sequences. On the attack side, researchers have analyzed inference under correlated releases and proposed empirical auditing of metric-aware guarantees Qiu et al. (2022); Yadav et al. (2024).

Although mDP brings needed geometry, it is framed as bounds on output-distribution ratios for pairs of inputs and is usually analyzed per release, whereas real adversaries often act on *posterior beliefs* and routinely aggregate *correlated outputs* (e.g., multiple perturbed locations within a trajectory Yadav et al. (2024) or multiple words within a sentence Staab et al. (2024)). These observations motivate a shift in how we evaluate and enforce metric privacy. Rather than bounding per-release output-distribution ratios in isolation, we ask how an attacker's posterior beliefs sharpen when multiple, correlated releases are consumed jointly, especially by learned models that excel at aggregating such evidence.

## 1.1 OUR WORK

*We rethink metric privacy through metric-normalized posterior leakage (mPL)*, an attacker-aligned, geometry-aware criterion that natively handles *joint* (set/sequence) consumption and is *auditable* with learned adversaries (**Contribution 1**). We formalize mPL (*Definition 2*) as a distance-calibrated change in posterior odds between candidate secrets after observing releases, and define *bounded mPL* by constraining mPL to a budget $\epsilon$ (*Definition 3*). We then establish standard stability properties—most notably *post-processing invariance* (*Proposition 1*)—and prove that, for single or independent releases, uniformly bounding mPL is equivalent to mDP (*Propositions 2–3*).

Next, we study joint observation of multiple perturbed records with correlated secrets (**Contribution 2**). We show that, even if a perturbation mechanism satisfies per-record mDP, it can still fail to ensure mPL under joint observation. We demonstrate this both when the joint secret record distribution is specified explicitly and when it is implicit and learned by neural adversaries. In particular, using three *deep neural network (DNN)* adversaries, *recurrent neural network (RNN)*, *long short-term memory (LSTM)*, and *Transformer*, we find that standard perturbation methods such as the *exponential mechanism (EM)* (which satisfies mDP) still incur mPL violations of 12.0%, 11.1%, and 14.4%, respectively.

As a countermeasure, we adopt a *trust-and-verify* pipeline, called *Adaptive mPL (AmPL)*, that places a learned attacker in the loop (**Contribution 3**). In each round, we (i) apply level-wise perturbation, (ii) train high-capacity adversaries, such as RNN, LSTM, and Transformer, on jointly observed releases to infer secrets and estimate mPL, (iii) update perturbation strengths using the adversary's feedback to balance privacy and utility, and (iv) optionally perform *Bayesian remapping* Chatzikokolakis et al. (2017) as post-processing to recover utility without weakening privacy guarantees. Considering universal worst-case control under dependence is onerous, we introduce a *probabilistically bounded* variant, *PBmPL (Definition 3)*, which limits how often mPL may exceed a target budget. Concretely, we evaluate leakage on sampled inputs and releases, measure the fraction of instances that violate the budget, and require this empirical violation rate to stay below a chosen tolerance with appropriate confidence.

Finally, we conduct a case study on text embeddings (**Contribution 4**), where *personally identifiable information (PII)* and *potentially identifying information (PoII)* are perturbed under the proposed framework. Our results show that standard mDP mechanism, like expoenential mechanism, may still lead to high posterior leakage (e.g., Transformer adversary yields $\approx 0.33$ leakage under mDP), violating the mPL requirement. In contrast, AmPL substantially reduces leakage (down to $\approx 0.12$), while preserving similar utility. This case study illustrates how AmPL can be applied in practice and highlights the trade-off between semantic utility and privacy protection.

## 2 METRIC-NORMALIZED POSTERIOR LEAKAGE

In this section, we introduce *mPL* and analyze its properties. We begin with preliminaries (§2.1), including the secret and perturbation spaces, the threat model, and mDP. We then formalize mPL (§2.2) and show that, for single (or independent) releases, uniformly bounding mPL is equivalent to mDP (§2.3). Finally, we extend to joint observations with correlated secrets, illustrating why per-record mDP can be insufficient to guarantee mPL (§2.4). The main notations used throughout the paper are summarized in Table 2 (Appendix B).

### 2.1 PRELIMINARIES

*Data perturbation* is a widely used privacy-preserving technique that protects sensitive information by deliberately adding noise to the original data before releasing individual records. Formally, a perturbation mechanism $\mathcal{M}$ is defined as a probabilistic mapping $\mathcal{M} : \mathcal{X} \to \mathcal{Y}$, where $\mathcal{X}$ is the domain of secret records and $\mathcal{Y}$ is the domain of perturbed records. Distances between records in $\mathcal{X}$ are measured by a function $d : \mathcal{X}^2 \to \mathbb{R}_+$, with $d_{x_i,x_j}$ denoting the distance between any pair $x_i, x_j \in \mathcal{X}$. We consider a setting with $L$ jointly observed secret records, denoted by $\mathbf{x} = (x^1, \ldots, x^L) \in \mathcal{X}^L$. For each $\ell \in [L]$, let $X^\ell$ and $Y^\ell$ denote the random variables corresponding to the $\ell$-th secret record and its perturbed counterpart, respectively.

**Definition 1** (mDP). *Let $(\mathcal{X}, d)$ be a metric secret space and let $\mathcal{M}$ be a perturbation mechanism with input space $\mathcal{X}$ and output space $\mathcal{Y}$. We say that $\mathcal{M}$ satisfies $(\epsilon, d)$-mDP if,*

$$\sup_{x_i \neq x_j} \sup_{y \in \mathcal{Y}} \ln \frac{\Pr[\mathcal{M}(x_i) = y]}{\Pr[\mathcal{M}(x_j) = y]} \leq \epsilon d_{x_i,x_j}. \tag{1}$$

*where $\epsilon$ denotes the* privacy budget.

Intuitively, mDP requires that small changes in input $x$ induce only bounded changes in the law of $\mathcal{M}(x)$, yielding privacy calibrated to the metric $d$. A smaller $\epsilon$ implies a tighter bound—and hence stronger privacy—so that less can be inferred about $x$ from observing $\mathcal{M}(x)$.

Following Wang et al. (2017a); Liu & Qiu (2025); Imola et al. (2022), we consider a discrete perturbation space $\mathcal{Y} = \{y_1, \ldots, y_K\}$. To facilitate analysis, we represent $\mathcal{M}$ as a deterministic function $\tilde{\mathcal{M}}$ defined by $\mathcal{M}(x) \equiv \tilde{\mathcal{M}}(x, Z)$, where $Z \sim \mathrm{Uniform}(0, 1)$ is an auxiliary random variable that captures the randomness of $\mathcal{M}$. Specifically, we define cumulative sums

$$F_0(x) = 0, \qquad F_k(x) = \sum_{v=1}^{k} \Pr[\mathcal{M}(x) = y_v], \quad k = 1, \ldots, K. \tag{2}$$

Then $\tilde{\mathcal{M}}$ is given by $\tilde{\mathcal{M}}(x, Z) = \sum_{k=1}^{K} y_k \, \mathbf{1}_{[F_{k-1}(x), F_k(x))}(Z)$, where $\mathbf{1}_{[a,b)}(Z)$ is the indicator function, equal to 1 if $Z \in [a, b)$ and 0 otherwise.

In the following, we use both notations $\mathcal{M}$ and $\tilde{\mathcal{M}}$: $\mathcal{M}$ for simplicity of exposition, and $\tilde{\mathcal{M}}$ in formal arguments (e.g., the proof of Proposition 3).

**Adversarial model.** We adopt standard security assumptions commonly used in the literature on privacy-preserving computing Liu & Qiu (2025). The server is assumed to be *honest-but-curious*: it correctly follows the prescribed protocol but may attempt to infer sensitive information about individual users from the data it receives. We consider a prior-informed attacker whose goal is to identify the true secret $x_\ell$ from a candidate set $\mathcal{X}_\ell$ given one or more noisy releases. The attacker knows the perturbation mechanism $\mathcal{M}$ and its parameters, has access to an auxiliary corpus from the same population (to approximate the prior and train a posterior estimator), and can passively aggregate multiple correlated releases for the same secret. Given at least a single perturbed record $y$ and the mechanism $\mathcal{M}$, the server can infer the posterior distribution of $X$ using Bayes' rule,

$$\Pr(X = x \mid \mathcal{M}(X) = y) = \frac{\Pr(\mathcal{M}(X) = y \mid X = x)\Pr(X = x)}{\sum_{x' \in \mathcal{X}} \Pr(\mathcal{M}(X) = y \mid X = x')\Pr(X = x')}. \tag{3}$$

Such adversarial model captures the realistic scenario in which the server is operated by an organization with incentives to collect and analyze data. Therefore, the server cannot be fully trusted from a privacy perspective. On the user side, all users are assumed to be honest and faithfully execute the perturbation protocol before sending their data to the server. We do not consider collusion between users or between users and the server, as such settings typically fall outside the scope of LDP/mDP-based frameworks and require alternative threat models To et al. (2017).

**Joint observation.** In this paper, we consider attackers who observe *multiple* perturbed releases. Given a user's secret sequence $\mathbf{x} = (x^{(1)}, \ldots, x^{(L)})$, a randomized mechanism $\mathcal{M}$ is applied *independently* to each component, producing $\mathbf{y} = (y^{(1)}, \ldots, y^{(L)})$ with $y^{(\ell)} = \mathcal{M}(x^{(\ell)})$ for $\ell \in [L]$. The adversary observes the joint output $\mathbf{y}$ and aims to infer the original secrets $\mathbf{x}$. Importantly, while the perturbations are applied independently across $\ell$, the secrets $\{x^{(\ell)}\}_{\ell=1}^{L}$ may be statistically dependent. Then, the adversary estimates the posterior distribution of each secret record $X_\ell$ conditioned on the full perturbed sequence $\mathbf{y}$ as:

$$\Pr[X_\ell = x \mid \{\mathcal{M}(X_1), \ldots, \mathcal{M}(X_L)\} = \mathbf{y}], \quad x \in \mathcal{X}. \tag{4}$$

## 2.2 Formalization of mPL and Its Post-Processing Property

To quantify how much the joint observation $\mathbf{y}$ reveals about the original input $\mathbf{x}$, we define *mPL* as the change in relative likelihood (posterior odds) between two candidate records $x_i$ and $x_j$ from prior to posterior after observing $\mathbf{y}$.

**Definition 2** (mPL). mPL *between a pair of records* $x_i, x_j \in \mathcal{X}$ *given the joint observation* $\mathbf{y} = (y^1, \ldots, y^L)$ *is defined as*

$$\mathrm{mPL}_{\mathcal{M}}(x_i, x_j, \mathbf{y}) = \frac{1}{d_{x_i, x_j}} \left| \ln \frac{\Pr(X_\ell = x_i \mid \{\mathcal{M}(X_1), \ldots, \mathcal{M}(X_L)\} = \mathbf{y})}{\Pr(X_\ell = x_j \mid \{\mathcal{M}(X_1), \ldots, \mathcal{M}(X_L)\} = \mathbf{y})} - \ln \frac{\Pr(X_\ell = x_i)}{\Pr(X_\ell = x_j)} \right|. \tag{5}$$

Here, the prior ratio $\frac{\Pr(X_\ell = x_i)}{\Pr(X_\ell = x_j)}$ reflects the likelihood of $X_\ell$ being $x_i$ versus $x_j$ before any observation, while the posterior ratio $\frac{\Pr(X_\ell = x_i \mid \{\mathcal{M}(X_1), \ldots, \mathcal{M}(X_L)\} = \mathbf{y})}{\Pr(X_\ell = x_j \mid \{\mathcal{M}(X_1), \ldots, \mathcal{M}(X_L)\} = \mathbf{y})}$ captures the updated belief after observing $y$. *The posterior leakage* $\mathrm{mPL}_{\mathcal{M}}(x_i, x_j, \mathbf{y})$ *thus measures the change in these relative beliefs,*

*normalized by the record distance $d_{x_i,x_j}$. A smaller leakage value indicates that the perturbation mechanism $\mathcal{M}$ reveals less information, thereby offering stronger privacy protection.*

Notably, the key distinction between the new posterior inference model (as defined in Eq. (4)) and the existing posterior inference model Kifer & Machanavajjhala (2012) (as defined in Eq. (3)) lies in *the attacker inferring each $X_\ell$ using the joint observation of $\mathbf{y}$ instead of the observation of individual perturbed record $y^\ell$.*

**Definition 3** ($\epsilon$-Bounded mPL). *A randomized perturbation mechanism $\mathcal{M}$ is said to satisfy $\epsilon$-bounded mPL if, for every pair of distinct secrets $x_i \neq x_j$ and every joint observation $\mathbf{y} \in \mathcal{Y}^L$,*

$$\sup_{x_i \neq x_j} \sup_{\mathbf{y} \in \mathcal{Y}^L} \mathrm{mPL}_{\mathcal{M}}(x_i, x_j, \mathbf{y}) \leq \epsilon. \tag{6}$$

**Proposition 1** (Post-processing for bounded mPL). *Suppose the data perturbation mechanism $\mathcal{M}$ satisfies the $\epsilon$-bounded joint mPL constraint: $\sup_{x_i \neq x_j} \sup_{\mathbf{y} \in \mathcal{Y}^L} \mathrm{mPL}_{\mathcal{M}}(x_i, x_j, \mathbf{y}) \leq \epsilon$. Then for any function $f$, the post-processed output $f \circ \mathcal{M}$ also satisfies the bounded joint mPL constraint:*

$$\sup_{x_i \neq x_j} \sup_{\mathbf{y} \in \mathcal{Y}^L} \mathrm{mPL}_{f \circ \mathcal{M}}(x_i, x_j, \mathbf{z}) \leq \epsilon. \tag{7}$$

*where $\mathcal{Z} = \mathrm{Range}(f \circ \mathcal{M})$. Detailed proof can be found in Appendix D.1.*

Intuitively, Proposition 1 implies that, if for every possible perturbed records $\mathbf{y}$, their joint mPL is within the privacy budget $\epsilon$, then post-processing the output using $f(\mathbf{y})$ cannot amplify this ratio, since post-processing coarsens the output space, mixing outcomes, which cannot increase the distinction between $X_\ell = x_i$ and $X_\ell = x_j$.

## 2.3 Properties Based on Individual or Independent Observations

**Proposition 2** (Single-observation equivalence of mPL and mDP). *Let $\mathcal{M}$ be a perturbation mechanism on a metric secret space $(\mathcal{X}, d)$. Define the single-observation mPL for a pair $(x_i, x_j)$ and observation $y$ by*

$$\mathrm{mPL}_{\mathcal{M}}((x_i, x_j), y) = \frac{1}{d_{x_i,x_j}} \left| \ln \frac{\Pr(X = x_i \mid \mathcal{M}(X) = y)}{\Pr(X = x_j \mid \mathcal{M}(X) = y)} - \ln \frac{\Pr(X = x_i)}{\Pr(X = x_j)} \right|. \tag{8}$$

*For any $\epsilon \geq 0$, $\mathcal{M}$ satisfies $(\epsilon, d)$-mDP if and only if the single-observation mPL bound holds, i.e., $\sup_{x_i \neq x_j} \sup_{y \in \mathcal{Y}} \mathrm{mPL}_{\mathcal{M}}((x_i, x_j), y) \leq \epsilon$. A detailed proof appears in Appendix D.2.*

While real-world record often contains dependencies (e.g., between a person's name and organization), analyzing posterior leakage under the simplifying assumption that protected records $X_1, \ldots, X_L$ are *independently distributed* offers useful theoretical insights. Under this assumption, we establish a connection between individual and joint posterior leakage in *Proposition 3*:

**Proposition 3** (Independent-observation equivalence of mPL and mDP). *If the $L$ secret words $X_1, \ldots, X_L$ are independently distributed, then ensuring $\sup_{x_i \neq x_j} \sup_{y^\ell \in \mathcal{Y}} \mathrm{mPL}_{\mathcal{M}}((x_i, x_j), y^\ell) \leq \epsilon$ for each $y^\ell$ ($\ell = 1, ..., L$) is sufficient to guarantee $\sup_{x_i \neq x_j} \sup_{\mathbf{y} \in \mathcal{Y}^L} \mathrm{mPL}_{\mathcal{M}}(x_i, x_j, \mathbf{y}) \leq \epsilon$. A detailed proof appears in Appendix D.3.*

The proposition shows that without inter-token dependencies, individual-level mDP bounds suffice to ensure privacy under joint observation. However, this assuption rarely holds in practice.

## 2.4 Threat Models Based on Joint and Correlated Observations

In this part, we relax the independence assumption and introduce more realistic threat models where the records $X_1, \ldots, X_L$ are dependent.

**(1) Explicit joint-probability attacker (a toy example).** We consider an attacker that models the *joint* distribution of two secrets and performs *Bayesian inference* over two perturbed outputs. Let $X_1, X_2 \in \mathcal{X} = \{x_1, x_2\}$ with a correlated prior: $\Pr(X_1 = x_1, X_2 = x_1) = \Pr(X_1 = x_2, X_2 = x_2) = 0.01$ and $\Pr(X_1 = x_1, X_2 = x_2) = \Pr(X_1 = x_2, X_2 = x_1) = 0.49$, and set $\epsilon = 1.0$.

For an exponential mechanism (EM) perturbation $\mathcal{M}_{\mathrm{EM}}$ with two outputs $\{y_1, y_2\}$, suppose $\Pr(\mathcal{M}_{\mathrm{EM}}(X_i) = y_1 \mid X_i = x_1) = 0.72$, $\Pr(\mathcal{M}_{\mathrm{EM}}(X_i) = y_2 \mid X_i = x_1) = 0.28$, $\Pr(\mathcal{M}_{\mathrm{EM}}(X_i) = y_1 \mid X_i = x_2) = 0.28$, $\Pr(\mathcal{M}_{\mathrm{EM}}(X_i) = y_2 \mid X_i = x_2) = 0.72$. A direct calculation shows that for each $y_k \in \{y_1, y_2\}$,

$$\mathrm{mPL}_{\mathcal{M}_{\mathrm{EM}}}(x_1, x_2, y_k) = 0.944 < \epsilon, \tag{9}$$

so observing each perturbed record *individually* does not violate the mPL bound (therefore also achieving mDP according to Proposition 4).

In contrast, when the two outputs are consumed *jointly*, we obtain

$$\text{mPL}_{\mathcal{M}_{\text{EM}}}\big(x_1, x_2, \{\mathcal{M}_{\text{EM}}(x_1), \mathcal{M}_{\text{EM}}(x_2)\} = \{y_1, y_2\}\big) = 1.846 > \epsilon, \tag{10}$$

demonstrating that joint consumption under a correlated prior can trigger posterior-leakage violations even when all single-observation checks pass. Full details appear in Appendix C.1.

**(2) Inference models based on implicit joint probability.** Explicit Bayesian joint inference can, in principle, expose joint leakage, but it is often impractical: exact posteriors require normalizing over exponentially many configurations in sequence length (the denominator scales like $|\mathcal{X}|^N$), and the true joint probability distribution is unknown or misspecified.

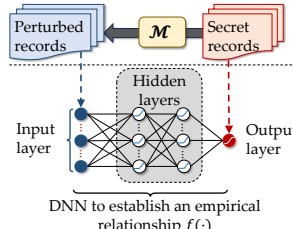

In these models, we instantiate the attacker as a high-capacity neural posterior estimator that directly approximates $\Pr(X_\ell = x_i | \{\mathcal{M}(X_1), \ldots, \mathcal{M}(X_L)\} = \mathbf{y})$. Specifically, we employ three DNN architectures, *RNN*, *LSTM*, and *Transformer*, to reconstruct secret records from their perturbed counterparts. As illustrated in Fig. 1, we apply the perturbation mechanism $\mathcal{M}$ to all the text and generate corresponding perturbed records. Then we randomly select 80% of secret–perturbed pairs, 60% serving as samples for training and 20% for validation, with each model minimizing the *mean squared error (MSE)* between the predicted and true records. We use the Adam optimizer Kingma & Ba (2017) with an initial learning rate of 0.001, reducing it when validation performance plateaus.

Figure 1: DNN-based Threat Model.

*Posterior approximation.* Given the estimated secret record $\hat{x} \in \mathcal{X}$, we first compute the squared Euclidean distance between $\hat{x}$ and each candidate secret record $x_i \in \mathcal{X}$: $d_{\hat{x}, x_i}^2 = \|\hat{x} - x_i\|_2^2$. We then convert these distances into a probability distribution using a temperature-scaled Gaussian softmax Guo et al. (2017): $\Pr(X = x_i \mid Y = y) = \frac{\exp(-d_{\hat{x}, x_i}^2 / (\tau_{\text{base}} \cdot \tau))}{\sum_{x_j \in \mathcal{X}} \exp(-d_{\hat{x}, x_j}^2 / (\tau_{\text{base}} \cdot \tau))}$, where $\tau_{\text{base}}$ and $\tau > 0$ are temperature parameters that jointly control the sharpness of the posterior distribution. This construction yields a numerically stable distribution that can be interpreted as an approximation of the posterior over hidden states $X$ given the observation $y$.

*Prior approximation.* When every sensitive token is deterministically replaced by a fixed placeholder $y_{\text{mask}}$ (e.g., ``xxxx'' for word embedding) rather than perturbed via $\mathcal{M}$, the observation contains no information about the underlying secret word. In this case, the likelihood is uniform across all candidates, and the posterior distribution reduces to the prior.

**Discussion: Per-user accounting.** When records can be cleanly grouped by user and the mechanism is designed with explicit per-user accounting, a per-user privacy budget is a natural way to mitigate composition across correlated records. Our focus, however, is on settings where such user grouping is unavailable or unreliable (as in many text/embedding applications and our public datasets, which lack user identifiers), so we adopt a user-agnostic formulation that targets joint leakage over arbitrary correlated secrets rather than per-user privacy loss. We provide a more detailed discussion of this design choice and how our framework can be combined with per-user budgeting when reliable user IDs are available in **Appendix C.2**.

## 3  DATA PERTURBATION FRAMEWORK: PERTURB−AUDIT−ADAPT

As analyzed in Section 2.4, an exact, closed-form calibration of mechanism parameters for mPL is generally intractable: the posterior needed to evaluate mPL depends on high-dimensional, correlation-aware likelihoods. To operationalize mPL, we therefore adopt a *trust–and–verify* framework with an attacker in the loop, called *Adaptive mPL (AmPL)*. AmPL *trusts* a candidate perturbation (initialized from a principled per-record design suggested by the independent case), trains a high-capacity adversary to approximate the posteriors on the resulting releases, and then *verifies* (and adapt) by auditing approximaed mPL derived by the learned attacker. Iterating this loop breaks the chicken-and-egg dependency, i.e., mechanism parameters need attacker feedback, and the attacker needs mechanism-perturbed data for training its inference model.

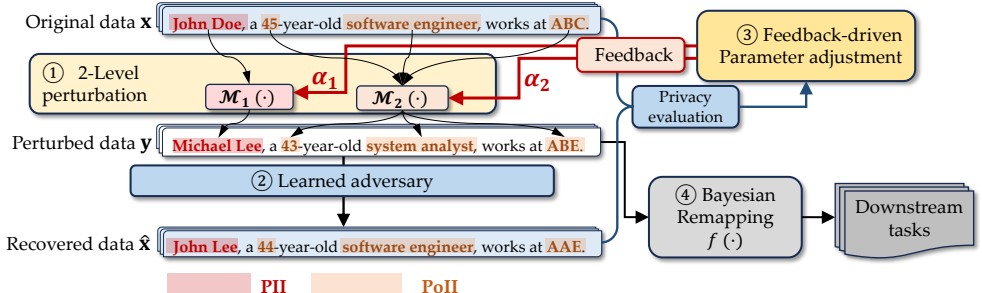

Figure 2: Illustration of the AmPL Framework (example: protecting PII and PoII word embeddings).

Additionally, AmPL stratifies secrets into two sensitivity tiers. *Personally identifiable information (PII)* refers to attributes that can directly identify or authenticate an individual, such as full name, email address, phone number, or precise home address. *Potentially identifying information (PoII)* denotes attributes that may not uniquely identify a person on their own, but can substantially reduce the anonymity set or reveal sensitive traits when combined with other data (e.g., employer, city of residence, demographic descriptors, or fine-grained preferences). Our tiered sensitivity model allows us to apply stricter privacy parameters to PII while still monitoring joint leakage for PoII.

Fig. 2 illustrates the AmPL framework using an example of protecting PII and PoII word embeddings. In each round, AmPL executes the following steps:

① **Level-wise data perturbation:** Let $\mathbf{x}$ denote the original data representation (e.g., PII and PoII word embeddings). We partition the whole secret record set $\mathcal{X}$ into $N$ tiers and apply perturbation mechanisms with $N$ levels, respectively. The perturbed representation is $\mathbf{y}$.

② **Adversarial DNN inference:** We train an adversarial model (e.g., RNN/LSTM/Transformer) to reconstruct $\mathbf{x}$ from $\mathbf{y}$ or to infer protected attributes, yielding $\hat{\mathbf{x}}$ (or predicted records). Using $\hat{\mathbf{x}}$, we evaluate the mPL violation ratio to quantify privacy risk under the adversary's inference.

③ **Feedback-driven adjustment:** The mPL violation ratio evaluation provides feedback to adapt the perturbation strength. We iteratively adjust the perturbation mechanisms to balance privacy protection and utility, converging toward an operating point that satisfies target leakage thresholds.

④ **Bayesian remapping:** To mitigate utility loss, we apply a Bayesian remapping function $f(\cdot)$ to $\mathbf{y}$ to obtain task-aligned representations for downstream use. Because $f$ is pure post-processing, it preserves the original privacy guarantees while improving task accuracy (according to *Proposition 1*).

Next, we introduce the details of Steps ①–④.

DETAILED STEPS

**Step ①: Level-wise data perturbation.** Let $N \in \mathbb{N}$ denote the number of sensitivity tiers. We first partition $\mathcal{X}$ into disjoint subsets $\{\mathcal{X}^{(1)}, \ldots, \mathcal{X}^{(N)}\}$ and define a level-assignment function $g : \mathcal{X} \to \{1, \ldots, N\}$ that maps each secret $x \in \mathcal{X}$ to its sensitivity level. For each level $\ell \in \{1, \ldots, N\}$, specify a mechanism $\mathcal{M}_\ell(\cdot; \alpha_\ell)$ with privacy/perturbation parameter $\alpha_\ell$. Given $x$, the released output is $y \sim \mathcal{M}_{g(x)}(x; \alpha_{g(x)})$, so that protection strength matches the sensitivity of $x$. The collection $\{\alpha_\ell\}_{\ell=1}^L$ can be tuned (e.g., via validation or feedback control) to meet target leakage–utility trade-offs.

*Two-level example for word-embedding privacy (PII vs. PoII):* In this case, we let $N = 2$ with $\mathcal{X}^{(1)}$ denoting direct identifiers (PII) and $\mathcal{X}^{(2)}$ denoting quasi-identifiers (PoII). We assign a stronger perturbation to PII and a milder one to PoII (e.g., mechanisms $\mathcal{M}_1, \mathcal{M}_2$ with $\epsilon_1 < \epsilon_2$), reflecting their different disclosure risks. A detailed design and evaluation of this two-level word-embedding perturbation appear in Section 4 (Case Study) and Appendix E.

**Step ②: Learned Adversary.** To evaluate and mitigate posterior leakage under realistic adversarial settings, we adopt a learned adversary approach based on DNNs, as introduced in Section 2.4. Specifically, models such as RNNs, LSTMs, and Transformers are trained to reconstruct the original records from their perturbed versions, effectively simulating strong inference attacks that exploit semantic dependencies across tokens. We then use the outputs of these adversarial models, i.e., the approximated posterior distributions over sensitive tokens, to assess whether the posterior leakage bounds are satisfied.

Notably, enforcing the posterior leakage constraint for *all* secret record pairs and perturbed records can lead to an overly conservative privacy budget. To address this, we adopt a probabilistic relaxation that requires the constraint to hold with high probability rather than deterministically.

**Definition 4** (Probabilistic bounded mPL). *Given a perturbation mechanism $\mathcal{M}_\ell$ ($\ell = 1, ..., L$), we define the violation probability $p_{\mathcal{X}_\ell^2}$ as the probablity that the posterior leakage exceeds the privacy budget $\epsilon$ for a randomly sampled pair of records and output:*

$$p_{\mathcal{X}_\ell^2} = \Pr\left[\mathrm{mPL}_{\mathcal{M}_\ell}(X_i, X_j, Y) > \epsilon\right], \tag{11}$$

*where $X_i$ and $X_j$ are drawn from the secret record domain $\mathcal{X}_\ell$. We say $\mathcal{M}_\ell$ can achieve $(\delta, \epsilon_\ell)$-PBmPL if $p_{\mathcal{X}_\ell^2} \leq \delta$.*

Directly computing $p_{\mathcal{X}_\ell^2}$ is computationally prohibitive, as it requires evaluating all $|\mathcal{X}_\ell|^2$ record pairs. In practice, where $|\mathcal{X}_\ell|$ may range from tens of thousands to over one hundred thousand records (e.g., 5,448 PIIs and 5,492 PoIIs in AG-News dataset Zhang et al. (2015)), exhaustive evaluation is intractable. To address this, we estimate $p_{\mathcal{X}_\ell^2}$ via *random sampling*. Specifically, we uniformly sample a subset $\mathcal{S}_\ell \subseteq \mathcal{X}_\ell^2 \times \mathcal{Y}$ consisting of $S_\ell$ triplets $(x_i, x_j, y)$, and define the empirical estimate:

$$\hat{p}_{\mathcal{S}_\ell} = \tfrac{1}{S_\ell} \sum_{(x_i, x_j, y) \in \mathcal{S}_\ell} \mathbf{1}\left(\mathrm{mPL}_{\mathcal{M}_\ell}(x_i, x_j, y) > \epsilon\right), \tag{12}$$

where $\mathbf{1}(\cdot)$ denotes the indicator function. Because $\mathcal{S}_\ell$ is sampled uniformly, $\hat{p}_{\mathcal{S}_\ell}$ is an *unbiased estimator* of $p_{\mathcal{X}_\ell^2}$, i.e., $\mathbb{E}[\hat{p}_{\mathcal{S}_\ell}] = p_{\mathcal{X}_\ell^2}$. We further establish the following concentration guarantee:

**Proposition 4.** *[Concentration Guarantee for Probabilistic mDP Sampling] If the empirical violation rate satisfies $\hat{p}_{\mathcal{S}_\ell} = \xi\delta$ for some constant $\xi < 1$, then $\Pr\left[p_{\mathcal{X}_\ell^2} \leq \delta\right] \geq 1 - 2\exp(-2S_\ell(1-\xi)^2\delta^2)$. The detailed proof can be found in **Appendix D.4**.*

Proposition 4 shows that as the sample size $S_\ell$ increases, the bound $2\exp(-2S_\ell(1-\xi)^2\delta^2)$ rapidly approaches zero, ensuring that $\Pr[p_{\mathcal{X}_\ell^2} \leq \delta]$ approaches one.

**Proposition 5** (Asymptotic faithfulness of the mPL audit). *[Asymptotic faithfulness of the mPL audit] Let $p(x \mid y)$ denote the true posterior and $q_\theta(x \mid y)$ the adversary trained on $n$ i.i.d. pairs $(x_i, y_i)$ by minimizing empirical conditional cross-entropy over a fixed neural-network class. Assume:*

    **A1**. *(Clipped posteriors) $\mathcal{X}$ is finite, and there exists $\gamma > 0$ such that $p(x \mid y) \geq \gamma$ and $q_\theta(x \mid y) \geq \gamma$ for all $x \in \mathcal{X}$ and all $y$ (implemented in practice by softmax clipping).*

    **A2**. *(Metric regularity) The metric satisfies $d(x_i, x_j) \geq d_{\min} > 0$ whenever $x_i \neq x_j$.*

*Then for any fixed pair of candidates $x_i, x_j$ there exists a constant $C > 0$ (depending only on $\gamma$, $d_{\min}$, and the candidate set) such that*

$$\mathbb{E}_Y\left[\left|\mathrm{mPL}(x_i, x_j; Y) - \widetilde{\mathrm{mPL}}(x_i, x_j; Y)\right|\right] \leq C\,n^{-\alpha/2} \tag{13}$$

*for all sufficiently large $n$, where $\mathrm{mPL}$ and $\widetilde{\mathrm{mPL}}$ denote the mPL computed using $p$ and $q_\theta$, respectively. The detailed proof can be found in **Appendix D.5**.*

*Interpretation.* As the adversary is trained on more data, its learned posterior $q_\theta(x \mid y)$ converges to the true posterior $p(x \mid y)$ in expected KL, and the mPL computed from $q_\theta$ converges to the true mPL at a polynomial rate. Thus our empirical mPL audit is asymptotically faithful to the true posterior-leakage risk.

**Scope of the audit.** Notably, auditing AmPL necessarily depends on the chosen learned adversary: with complex DNN-based attackers, computing the full posterior (and thus exact mPL) is computationally infeasible. Our aim is therefore not to provide a universal worst-case upper bound, but to design a flexible framework that adapts to different threat models by plugging in different adversary classes. Formally, mPL and PBmPL are defined for arbitrary adversaries, and for any fixed adversary class our sampling procedure enjoys a standard concentration guarantee: the empirical violation rate converges to the true PBmPL violation probability as the number of samples grows. In practice, we instantiate this with high-capacity neural estimators, which act as strong but necessarily approximate proxies for Bayes-optimal attackers; stronger or alternative adversaries can always be integrated into the same audit loop, potentially revealing additional violations.

Additionally, our analysis focuses on a single joint release observed by a fixed attacker, and we do not provide a universal composition theorem for repeated releases. Instead, we position our framework as an adaptive auditing tool for empirically controlling joint leakage in regimes where classical per-user composition is not directly applicable.

**Step ③: Feedback-Driven Perturbation Adjustment.** To balance privacy protection and utility preservation, we adopt an adaptive optimization strategy that iteratively updates the scaling factors $\alpha_1$ and $\alpha_2$ based on adversarial feedback. This adaptation is guided by minimizing a composite loss function $\mathcal{L}(\boldsymbol{\alpha})$, which jointly captures privacy leakage and utility degradation:

$$\mathcal{L}(\boldsymbol{\alpha}) = \lambda_1 \cdot \mathcal{L}_{\text{privacy}}(\boldsymbol{\alpha}) + \lambda_2 \cdot \mathcal{L}_{\text{utility}}(\boldsymbol{\alpha}), \tag{14}$$

where $\lambda_1, \lambda_2 > 0$ are trade-off coefficients that balance the two objectives.

The privacy loss term $\mathcal{L}_{\text{privacy}}(\boldsymbol{\alpha})$ is given by the empirical violation rate $\hat{p}_{\mathcal{S}}$, which estimates the probability that posterior leakage exceeds the privacy budget $\epsilon$ over a sampled set of input-output pairs. The utility loss term $\mathcal{L}_{\text{utility}}(\boldsymbol{\alpha})$ represents the expected semantic distortion caused by the perturbation:

$$\mathcal{L}_{\text{utility}}(\boldsymbol{\alpha}) = \sum_{\ell=1}^{L} \sum_{x \in \mathcal{X}_1} \sum_{y \in \mathcal{Y}} \pi_x \, c_{x,y} \, \Pr\left(\mathcal{M}_\ell(x; \alpha_\ell) = y\right),$$

where $\pi_x$ denotes the prior probability of $x$, and $c_{x,y}$ quantifies the utility loss incurred by reporting $y$ when the true input is $x$.

**Step ④: Bayesian Remapping.** While perturbation mechanisms protect privacy by injecting noise into sensitive data, the resulting outputs may not be optimal for downstream tasks due to semantic distortion. To mitigate this utility loss, we employ *Bayesian remapping*, a post-processing step that refines perturbed outputs based on their inferred posterior distributions. Given a perturbed record $y$, Bayesian remapping selects an alternative output that minimizes the expected utility loss under the posterior, formally defined as:

$$f(y) = \arg\min_{y' \in \mathcal{Y}} \sum_{\ell=1}^{L} \sum_{x \in \mathcal{X}_\ell} \underbrace{\Pr\left[X = x \mid \mathcal{M}_\ell(x; \alpha_\ell) = y\right]}_{\text{posterior of } x \text{ given perturbed record } y} c_{x,y'}. \tag{15}$$

Notably, Chatzikokolakis et al. (2017) has proved that this transformation preserves the original mDP guarantees under individual perturbed observations. We extend this result by formally proving in Proposition 1 that the joint posterior leakage (joint mPL) constraint is also preserved under post-processing.

# 4 CASE STUDY: DATA PROTECTION OF PII IN TEXT DATASETS

To evaluate the performance of AmPL, we use text embedding protection as our case study, since embeddings are a ubiquitous interface for real systems (search, chat, recommendation). Text naturally exhibits layered sensitivities (personally identifiable information (PII) vs. potentially identifiable information (PoII)), matching our level-wise perturbation design, and it offers well-defined adversaries (reconstruction, attribute inference) together with standardized utility benchmarks (classification, retrieval), enabling clear privacy–utility evaluation. While our experiments focus on textual embeddings, both the mPL joint-leakage notion and the AmPL repair framework are modality-agnostic: they only require a metric over secrets, a utility loss, and a learned attacker. In principle, the same pipeline can be instantiated for other modalities such as images, tabular data, or audio, which we leave as important directions for future work (a more detailed discussion can be found in **Appendix C.3**). Additonally, we focus on embedding-space perturbations in our experiments, but the mPL/AmPL framework is agnostic to whether noise is applied to embeddings or directly to text; we elaborate on this point and on extensions beyond text in **Appendix C.4**. Next, we briefly introduce the settings of this case study and additional details can be found in **Appendix E**.

**Datasets.** We choose to use the widely used pre-trained embedding model GloVe Pennington et al. (2014). We evaluate our framework on three widely used benchmark datasets:

① The *AG-News Dataset* Zhang et al. (2015) is a large-scale news classification benchmark containing 120,000 training and 7,600 test samples across four categories: World, Sports, Business, and Science/Technology. It is commonly used for text classification task.

② The *IMDB Review Dataset* Maas et al. (2011) is a benchmark for binary sentiment classification. It contains 50,000 reviews, evenly split between positive and negative labels (25,000 each).

③ The *Amazon Review Dataset* Zhang et al. (2015) contians millions of user-generated reviews across diverse domains such as electronics, books, and clothing. The full collection has 34,686,770 Amazon reviews from 6,643,669 users on 2,441,053 products.

**Data perturbation methods - Exponential Mechanism (EM).** We use EM-based methods to perturb data in this case study. Prior work observes that adding Laplace noise to word embeddings fares poorly in discrete vocabularies: the perturbed vector seldom corresponds to a valid token Carvalho et al. (2021). Nearest-neighbor postprocessing has been proposed to snap noisy vectors back to the vocabulary Feyisetan et al. (2019), but this (i) omits the privacy loss incurred by the search itself and (ii) ignores local density variations, often degrading utility under mDP. By contrast, EM directly *selects* a token from a finite candidate set using a distance-aligned utility score while preserving privacy guarantees by design, offering greater flexibility and typically better utility for non-numeric or discrete outputs Feyisetan et al. (2020a).

In particular, given the original token embedding set $\mathcal{X}$, we divide it into two levels: (1) PII, which directly identifies individuals, and (2) PoII, which may indirectly reveal identity. To protect them, we apply two *adjusted EM* perturbation mechanisms, $\mathcal{M}_{\mathrm{EM}}(\cdot; \alpha_1\epsilon)$ and $\mathcal{M}_{\mathrm{EM}}(\cdot; \alpha_2\epsilon)$, controlled by scaling factors $\alpha_1$ and $\alpha_2$ (with $\alpha_1, \alpha_2 \in [0, 1]$ and $\alpha_1 < \alpha_2$). Formally, for $\ell \in \{1, 2\}$,

$$\Pr[\mathcal{M}_{\mathrm{EM}}(x; \alpha_\ell\epsilon) = y] = \frac{\exp\left(-\frac{1}{2}\alpha_\ell\epsilon \cdot d_{x,y}\right)}{\sum_{y' \in \mathcal{Y}} \exp\left(-\frac{1}{2}\alpha_\ell\epsilon \cdot d_{x,y'}\right)}, \tag{16}$$

where $\mathcal{M}_1(\cdot)$ (with smaller $\alpha_1$) introduces stronger noise for PII, and $\mathcal{M}_2(\cdot)$ (with larger $\alpha_2$) applies milder noise for PoII.

**Compared methods.** As a baseline, we use the standard EM in Eq. (16) with no level differentiation, i.e., $\alpha_1 = \alpha_2 = 1$. For ablations, we compare our full method (AmPL) against two variants: (i) *AmPL-U* (utility-preserving), which removes identity-salience weighting by setting $\alpha_1 = 0$ in the objective of Eq. (14), thereby optimizing only utility loss; and (ii) *AmPL-1*, which applies identity-salience weighting to PII only (no PoII weighting).

Table 1: Posterior leakage violation ratio (%).

| | RNN | | | LSTM | | | Transformer | | |
|---|---|---|---|---|---|---|---|---|---|
| $\epsilon$ | 0.30 | 0.40 | 0.50 | 0.30 | 0.40 | 0.50 | 0.30 | 0.40 | 0.50 |
| AG News Dataset | | | | | | | | | |
| EM (mDP) | 6.62±1.33 | 2.73±0.72 | 1.18±0.51 | 4.11±0.75 | 1.39±0.42 | 0.53±0.15 | 6.68±1.18 | 2.96±1.26 | 1.24±0.62 |
| AmPL-U | 6.51±1.52 | 2.75±0.66 | 1.14±0.61 | 4.15±0.85 | 1.40±0.28 | 0.50±0.23 | 6.84±1.24 | 2.99±1.26 | 1.22±0.62 |
| AmPL-1 | 10.28±2.61 | 3.81±1.89 | 1.38±0.64 | 3.41±1.81 | 1.24±0.93 | 0.37±0.31 | 6.14±3.77 | 2.38±1.84 | 0.81±0.76 |
| AmPL | 5.78±1.27 | 2.16±0.74 | 0.80±0.59 | 3.44±0.84 | 1.14±0.32 | 0.30±0.12 | 6.00±1.70 | 2.44±1.05 | 0.96±0.28 |
| IMDB Review Dataset | | | | | | | | | |
| EM (mDP) | 6.29±2.82 | 2.68±1.35 | 1.22±0.80 | 0.08±0.10 | 0.01±0.01 | 0.00±0.01 | 13.48±6.34 | 5.64±4.25 | 2.34±2.02 |
| AmPL-U | 6.21±2.66 | 2.74±1.25 | 1.21±0.80 | 0.08±0.09 | 0.01±0.01 | 0.00±0.01 | 13.40±6.27 | 5.66±4.18 | 2.32±1.96 |
| AmPL-1 | 11.55±5.84 | 5.37±3.68 | 1.79±0.95 | 1.45±1.98 | 0.24±0.57 | 0.04±0.15 | 22.53±17.69 | 11.99±14.63 | 5.08±7.54 |
| AmPL | 5.54±2.34 | 2.24±1.58 | 0.89±0.87 | 0.04±0.05 | 0.00±0.01 | 0.00±0.00 | 13.00±5.66 | 5.23±3.54 | 1.99±1.81 |
| Amazon Review Dataset | | | | | | | | | |
| EM (mDP) | 8.81±2.04 | 3.76±1.52 | 2.05±0.90 | 6.02±1.18 | 2.60±1.31 | 1.18±0.72 | 10.74±2.98 | 5.27±2.14 | 2.61±1.35 |
| AmPL-U | 8.81±2.14 | 3.91±1.47 | 2.05±0.97 | 6.02±1.39 | 2.63±1.36 | 1.17±0.71 | 10.77±3.31 | 5.35±2.03 | 2.58±1.28 |
| AmPL-1 | 23.40±11.16 | 11.62±8.68 | 6.00±6.38 | 11.54±8.21 | 5.66±5.86 | 2.58±3.13 | 23.56±19.73 | 12.66±15.92 | 6.59±8.25 |
| AmPL | 8.15±2.25 | 3.51±1.39 | 1.65±1.36 | 5.34±1.66 | 2.18±1.28 | 0.97±1.08 | 10.31±3.74 | 4.93±2.31 | 2.30±1.28 |

MAIN RESULTS

**EM leakage under different attackers.** Table 1 compares the mPL violation ratio of different perturbation approaches across the three datasets (AG News, IMDB, Amazon) and inference models (RNN, LSTM, Transformer). From the table, we can observe that

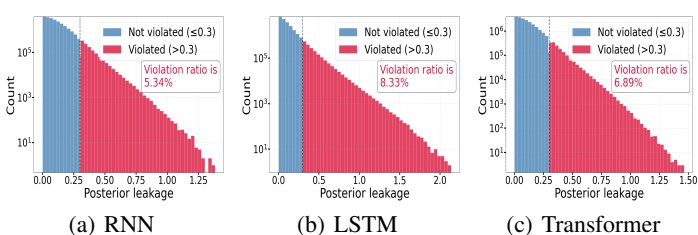

(a) RNN    (b) LSTM    (c) Transformer

Figure 3: Examples: mPL distribution derived by different DNN-based inference models.

even with per-record mDP noise (EM), mPL violations remain nontrivial under joint, learned attackers. For example, at $\epsilon = 0.30$ the Transformer attacker flags a sizeable fraction of violations

(e.g., $\approx 13.5\%$ on IMDB), and classical RNNs still expose leakage on AG NEWS (e.g., $\approx 6.6\%$). Figures 3(a)–(c) illustrate the distribution of mPL in the AG News dataset when $\epsilon = 0.30$.

We also observe that larger values of $\epsilon$ lead to lower violation rates. Recall that the empirical violation ratio is computed by checking whether the posterior odds exceed the bound $\exp(\epsilon d(s, s'))$. For small $\epsilon$, this bound is very tight, so even modest deviations caused by implicit correlations across multiple releases can trigger a violation. As $\epsilon$ increases, the allowable bound $\exp(\epsilon d(s, s'))$ grows faster than these correlation-induced deviations, so the same mechanism remains well within the feasible region and fewer pairs are counted as violations. While larger $\epsilon$ reduces these rates, the key takeaway is that per-record mDP alone does not control attacker-aligned leakage under joint consumption; stronger sequence models (Transformers) tend to reveal higher leakage than LSTMs.

According to the table, AmPL-U, which does not explicitly target leakage, performs similarly to EM, while AmPL-1 (without PoII protection) exhibits markedly higher violation rates, underscoring the importance of protecting quasi-identifiers. In contrast, AmPL consistently achieves the lowest violation ratios, demonstrating its robustness against adversarial inference. Overall, AmPL achieves roughly 5–15% reduction on average, with the largest relative gains on AG News (RNN/LSTM) and IMDB (LSTM), while maintaining the same low utility loss as shown in Fig. 4.

In our experiments, the mPL sampling sizes for `AG News`, `IMDB Reviews`, and `Amazon Reviews` are 11,214,777, 49,141,440, and 32,997,205, respectively. We set the achievable PBmPL target at $\tilde{\delta} = 1.05\,\delta^\star$ (i.e., 5% above the tightest observed violation $\delta^\star$ per dataset and $\epsilon$). Under this target, the PBmPL failure probabilities remain astronomically small across all settings: $\log_{10} p_{\text{fail}}$ ranges from roughly $-10^2$ (i.e., $10^{-100}$) down to below $-10^6$ (i.e., $< 10^{-1,000,000}$), indicating that violations are effectively impossible at scale. Detailed results are reported in **Appendix F.1**.

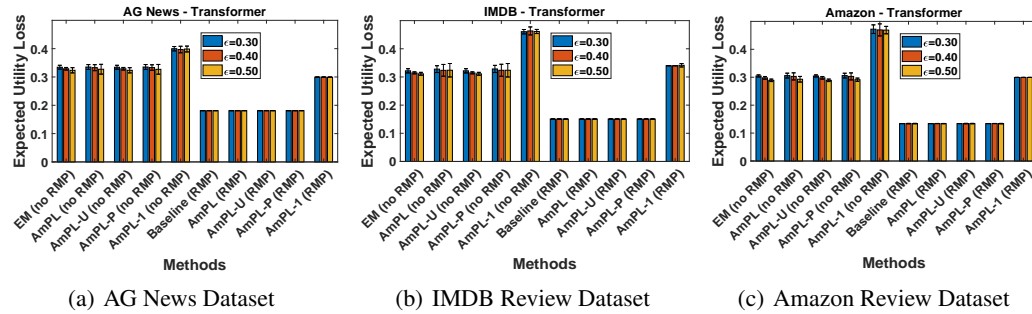

| (a) AG News Dataset | (b) IMDB Review Dataset | (c) Amazon Review Dataset |

Figure 4: Expected utility loss (applying Transformer as the adversarial model).

Fig. 4(a)(b)(c) compares the expected utility loss of the proposed perturbation mechanisms when the adversarial model applies Transformer (due to the limited space, the results of RNN and LSTM can be found in Appendix). From the figures, we observe that applying remap (RMP) significantly improves utility across all methods: the expected utility loss drops from around 0.32–0.33 to 0.18 on AG News, from 0.31–0.33 to 0.15 on IMDB, and from 0.29–0.31 to 0.13 on Amazon. *Notably, after remapping, the utility of EM, AmPL, and AmPL-U becomes nearly indistinguishable, with AmPL achieving almost the same utility loss as EM.* This occurs because the remap operation dominates the perturbation process by projecting perturbed embeddings back to utility-preserving regions, thereby neutralizing the fine-grained differences between these mechanisms. In contrast, AmPL-1 (RMP) remains less utility-preserving (e.g., $\approx 0.30$ on AG News and Amazon, $\approx 0.34$ on IMDB).

We also report learning curves for the learned adversary (**Appendix F.4**), showing that attack accuracy saturates quickly with a moderate number of training pairs, while the estimated mPL violation ratio slightly decreases as additional data yield better-calibrated posterior estimates.

## 5  CONCLUSIONS

We formalized metric-normalized posterior leakage and its high-probability relaxation (PBmPL), and proposed AmPL, an adaptive, attacker-aligned perturbation mechanism. Across AG News, IMDB, and Amazon with RNN/LSTM/Transformer, AmPL cuts posterior leakage while keeping utility loss low, yielding a favorable privacy–utility trade-off. Ablations show AmPL-U preserves performance with limited leakage reduction, whereas AmPL-1 boosts protection at modest extra cost. In the future work, we will extend beyond text, strengthen PBmPL composition, and scale to broader adversaries.

## 6 ETHICS STATEMENT

We adhere to the ICLR Code of Ethics and explicitly acknowledge this during submission. Our study uses publicly available text datasets and evaluates privacy leakage on PII/PoII under attacker models (RNN/LSTM/Transformer); no new human-subject data were collected and no individual-level interventions were performed. We minimize potential harm by (i) focusing on aggregate leakage metrics, (ii) releasing code that defaults to sanitized examples, and (iii) discussing dual-use risks and recommended safeguards (e.g., stricter thresholds, access controls). Any use of LLMs in writing or experiments is disclosed; authors remain fully responsible for all content.

## 7 REPRODUCIBILITY STATEMENT

In the supplementary file, we provide anonymized artifacts to ensure full reproducibility. Specifically, we include: complete training and evaluation scripts, data preprocessing pipelines, implementations of all three neural network architectures, utility loss computation modules, and configuration settings for all experimental variants. The code structure enables direct reproduction of all reported privacy ratios and utility losses results across the three benchmark datasets.

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

# Part I

# Appendix

## Table of Contents

## APPENDIX OVERVIEW

*Appendix A* discloses our use of large language models for wording and clarity. *Appendix B* consolidates mathematical notation (spaces, metrics, random variables, mechanisms, and posterior quantities). *Appendix C* develops inference under *explicit* joint probabilities, detailing Bayesian posteriors, normalization/tractability issues, and contrasts with learned (implicit) joint inference.  *Appendix D* contains omitted proofs, including post-processing invariance, single/independent equivalence between bounded mPL and mDP, and concentration bounds used for PBmPL. *Appendix E* provides additional case-study details (two-level perturbation for PII/PoII, parameterization, identity-salience weighting, and utility-loss computation). *Appendix F* reports extended experiments: recommended $\delta$ thresholds with failure-bound analysis across datasets and $\epsilon$ regimes, plus utility-loss curves when RNN/LSTM are used as adversaries to complement the Transformer results in the main text.

## A  THE USE OF LARGE LANGUAGE MODELS (LLMs)

We used a large language model (LLM) assistant to aid wording, improve clarity, and polish exposition across *Sections 1–7* and *Appendices B–F*. The LLM was not used to generate novel results, code, or citations, and no outputs were accepted without human review. The authors verified the accuracy of all assisted text and take full responsibility for the final content.

# B  MATH NOTATIONS

Table 2: Notation summary. Symbols are grouped by role (spaces/variables, mechanisms/metrics, leakage/certificates, and optimization).

| Symbol | Meaning |
|---|---|
| **Spaces and random variables** | |
| $\mathcal{X}, \mathcal{Y}$ | Secret (embedding) space and perturbed/release space. |
| $x_i \in \mathcal{X}, y_k \in \mathcal{Y}$ | A candidate secret embedding and a candidate perturbed embedding. |
| $X, X_\ell$ | Random variables for a secret embedding and the $\ell$-th secret in a sequence. |
| $d_{x_i, x_j}$ | Metric distance on $\mathcal{X}$ between $x_i$ and $x_j$. |
| $c_{x_i, y_k}$ | Utility loss when releasing $y_k$ for true input $x_i$. |
| **Mechanisms and parameters** | |
| $\mathcal{M}$ | Perturbation mechanism mapping $\mathcal{X} \to \mathcal{Y}$. |
| $\mathcal{M}(\cdot, \alpha_\ell)$ ($\ell = 1, ..., N$) | Level-wise mechanisms for $N$ sensitivity tiers (e.g., PII vs. PoII). |
| $\boldsymbol{\alpha} = (\alpha_1, \ldots, \alpha_N)$ | Vector of per-level perturbation strengths. |
| $f(y)$ | Bayesian remapping (post-processing) applied to releases. |
| **Leakage and certificates** | |
| $\mathrm{mPL}_{\mathcal{M}}(x_i, x_j; \mathbf{y})$ | Metric-normalized posterior leakage (single/joint; prior→posterior odds change, normalized by $d_{x_i, x_j}$). |
| $\epsilon$ | Target mPL budget (smaller is more private). |
| $p_{\mathcal{X}^2}$ | Violation probability $\Pr[\mathrm{mPL} > \epsilon]$ over random pairs $(x_i, x_j)$ and releases. |
| $\hat{p}_S$ | Empirical estimate of $p_{\mathcal{X}^2}$ from a sampled set $S$ of triples $(x_i, x_j, y)$. |
| $\delta$ | Tolerance on violation frequency for PBmPL. |
| **Optimization and adaptation (AmPL)** | |
| $L(\boldsymbol{\alpha})$ | Composite objective balancing privacy and utility. |
| $L_{\mathrm{privacy}}(\boldsymbol{\alpha})$ | Privacy term (e.g., empirical violation rate). |
| $L_{\mathrm{utility}}(\boldsymbol{\alpha})$ | Expected utility distortion under $\mathcal{M}$. |
| $\lambda_1, \lambda_2$ | Weights trading off privacy vs. utility in $L(\cdot)$. |
| $\eta(t)$ | Learning rate at iteration $t$ for adaptive updates. |
| $\|\boldsymbol{\alpha}^{(t+1)} - \boldsymbol{\alpha}^{(t)}\|_2$ | Step size between consecutive parameter updates. |

# C  DISCUSSIONS

## C.1  INFERENCE MODELS BASED ON EXPLICIT JOINT PROBOBALITY

We construct a scenario, in which (1) two secret records $X_1$ and $X_2$ are not independently distributed, (2) observing each perturbed record ($\mathcal{M}_{\mathrm{EM}}(X_1)$ or $\mathcal{M}_{\mathrm{EM}}(X_2)$) *individually* doesn't violate the posterior leakage bound, yet (3) observing $\mathcal{M}_{\mathrm{EM}}(X_1)$ and $\mathcal{M}_{\mathrm{EM}}(X_2)$ *jointly* causes a posterior leakage bound violation for $X_1$.

Suppose that $X_1$ and $X_2$ each take values in $\mathcal{X} = \{x_1, x_2\}$ with the following joint distribution:

$$\Pr(X_1 = x_1, X_2 = x_1) = 0.01, \quad \Pr(X_1 = x_1, X_2 = x_2) = 0.49, \tag{17}$$
$$\Pr(X_1 = x_2, X_2 = x_1) = 0.49, \quad \Pr(X_1 = x_2, X_2 = x_2) = 0.01. \tag{18}$$

Then each $X_i$ has marginal distribution: $\Pr(X_i = x_1) = 0.5$ and $\Pr(X_i = x_2) = 0.5$. Therefore,

$$\Pr(X_1 = x_i, X_2 = x_j) \neq \Pr(X_1 = x_i)\Pr(X_2 = x_j), \tag{19}$$

$\forall x_i, x_j \in \mathcal{X}$ indicating $X_1$ and $X_2$ are not independent.

We let $\mathcal{M}_{\mathrm{EM}}(X_i) \in \{y_1, y_2\}$. The perturbation probabilities are given by

$$\Pr\left(\mathcal{M}_{\mathrm{EM}}(X_i) = y_1 \mid X_i = x_1\right) = 0.72, \tag{20}$$

$$\Pr\left(\mathcal{M}_{\mathrm{EM}}(X_i) = y_2 \mid X_i = x_1\right) = 0.28, \tag{21}$$

$$\Pr\left(\mathcal{M}_{\mathrm{EM}}(X_i) = y_1 \mid X_i = x_2\right) = 0.28, \tag{22}$$

$$\Pr\left(\mathcal{M}_{\mathrm{EM}}(X_i) = y_2 \mid X_i = x_2\right) = 0.72. \tag{23}$$

Finally, we set the privacy budget $\epsilon = 1$.

**(1) The posterior by observing each individual perturbed record**: The *posterior odds* given the observation $\mathcal{M}_{\mathrm{EM}}(X_i) = y_1$ is

$$\left| \ln\left( \frac{\Pr(X_i = x_1 \mid \mathcal{M}_{\mathrm{EM}}(X_i) = y_1)}{\Pr(X_i = x_2 \mid \mathcal{M}_{\mathrm{EM}}(X_i) = y_1)} \bigg/ \frac{\Pr(X_i = x_1)}{\Pr(X_i = x_2)} \right) \right| \tag{24}$$

$$= \left| \ln\left( \frac{\Pr(\mathcal{M}_{\mathrm{EM}}(X_i) = y_1 \mid X_i = x_1)}{\Pr(\mathcal{M}_{\mathrm{EM}}(X_i) = y_1 \mid X_i = x_2)} \right) \right| \tag{25}$$

$$= \left| \ln\left( \frac{0.72}{0.28} \right) \right| \tag{26}$$

$$= 0.9444 \tag{27}$$

$$< \epsilon. \tag{28}$$

Similarly, the *posterior odds* given the observation $\mathcal{M}_{\mathrm{EM}}(X_i) = y_2$ is

$$\left| \ln\left( \frac{\Pr(X_i = x_1 \mid \mathcal{M}_{\mathrm{EM}}(X_i) = y_2)}{\Pr(X_i = x_2 \mid \mathcal{M}_{\mathrm{EM}}(X_i) = y_2)} \bigg/ \frac{\Pr(X_i = x_1)}{\Pr(X_i = x_2)} \right) \right| \tag{29}$$

$$= \left| \ln\left( \frac{\Pr(\mathcal{M}_{\mathrm{EM}}(X_i) = y_2 \mid X_i = x_1)}{\Pr(\mathcal{M}_{\mathrm{EM}}(X_i) = y_2 \mid X_i = x_2)} \right) \right| \tag{30}$$

$$= \left| \ln\left( \frac{0.28}{0.72} \right) \right| \tag{31}$$

$$= 0.9444 \tag{32}$$

$$< \epsilon. \tag{33}$$

indicating that $\left| \ln\left( \frac{\Pr(X_i = x_1 \mid \mathcal{M}_{\mathrm{EM}}(X_i) = y_j)}{\Pr(X_i = x_2 \mid \mathcal{M}_{\mathrm{EM}}(X_i) = y_j)} \big/ \frac{\Pr(X_i = x_1)}{\Pr(X_i = x_2)} \right) \right| \leq 1$ for each $X_i$ and $y_k$.

**(2) Posterior leakage give joint observation**: The *posterior odds* given the joint observation $\mathcal{M}_{\mathrm{EM}}(X_1) = y_1, \mathcal{M}_{\mathrm{EM}}(X_2) = y_2$ is

$$\left| \ln\left( \frac{\Pr(X_1 = x_1 \mid \mathcal{M}_{\mathrm{EM}}(X_1) = y_1, \mathcal{M}_{\mathrm{EM}}(X_2) = y_2)}{\Pr(X_1 = x_2 \mid \mathcal{M}_{\mathrm{EM}}(X_1) = y_1, \mathcal{M}_{\mathrm{EM}}(X_2) = y_2)} \bigg/ \frac{\Pr(X_1 = x_1)}{\Pr(X_1 = x_2)} \right) \right| \tag{34}$$

$$= \left| \ln\left( \frac{\Pr(X_1 = x_1 \mid \mathcal{M}_{\mathrm{EM}}(X_1) = y_1, \mathcal{M}_{\mathrm{EM}}(X_2) = y_2)}{\Pr(X_1 = x_2 \mid \mathcal{M}_{\mathrm{EM}}(X_1) = y_1, \mathcal{M}_{\mathrm{EM}}(X_2) = y_2)} \bigg/ 1 \right) \right| \tag{35}$$

$$= \left| \ln\left( \frac{\Pr(X_1 = x_1, \mathcal{M}_{\mathrm{EM}}(X_1) = y_1, \mathcal{M}_{\mathrm{EM}}(X_2) = y_2)}{\Pr(X_1 = x_2, \mathcal{M}_{\mathrm{EM}}(X_1) = y_1, \mathcal{M}_{\mathrm{EM}}(X_2) = y_2)} \right) \right| \tag{36}$$

$$= \left| \ln\left( \frac{\sum_{x \in \{x_1, x_2\}} \Pr(X_1 = x_1, X_2 = x, \mathcal{M}_{\mathrm{EM}}(X_1) = y_1, \mathcal{M}_{\mathrm{EM}}(X_2) = y_2)}{\sum_{x \in \{x_1, x_2\}} \Pr(X_1 = x_2, X_2 = x, \mathcal{M}_{\mathrm{EM}}(X_1) = y_1, \mathcal{M}_{\mathrm{EM}}(X_2) = y_2)} \right) \right| \tag{37}$$

$$= \left| \ln\left( \frac{\sum_{x \in \{x_1, x_2\}} \Pr(X_1 = x_1, X_2 = x) \Pr(\mathcal{M}_{\mathrm{EM}}(X_1) = y_1, \mathcal{M}_{\mathrm{EM}}(X_2) = y_2 \mid X_1 = x_1, X_2 = x)}{\sum_{x \in \{x_1, x_2\}} \Pr(X_1 = x_2, X_2 = x) \Pr(\mathcal{M}_{\mathrm{EM}}(X_1) = y_1, \mathcal{M}_{\mathrm{EM}}(X_2) = y_2 \mid X_1 = x_2, X_2 = x)} \right) \right|$$

$$= \left| \ln\left( \frac{\sum_{x \in \{x_1, x_2\}} \Pr(X_1 = x_1, X_2 = x) \Pr(\mathcal{M}_{\mathrm{EM}}(X_1) = y_1 \mid X_1 = x_1) \Pr(\mathcal{M}_{\mathrm{EM}}(X_2) = y_2 \mid X_2 = x)}{\sum_{x \in \{x_1, x_2\}} \Pr(X_1 = x_2, X_2 = x) \Pr(\mathcal{M}_{\mathrm{EM}}(X_1) = y_1 \mid X_1 = x_2) \Pr(\mathcal{M}_{\mathrm{EM}}(X_2) = y_2 \mid X_2 = x)} \right) \right|$$

$$= \left| \ln\left( \frac{0.01 \times 0.72 \times 0.28 + 0.49 \times 0.72 \times 0.72}{0.49 \times 0.28 \times 0.28 + 0.01 \times 0.28 \times 0.72} \right) \right| \tag{38}$$

$$= 1.8456 \tag{39}$$

$$> \epsilon. \tag{40}$$

$$\left| \ln\left( \frac{\Pr(X_1 = x_1 \mid \mathcal{M}_{\mathrm{EM}}(X_1) = y_1, \mathcal{M}_{\mathrm{EM}}(X_2) = y_2)}{\Pr(X_1 = x_2 \mid \mathcal{M}_{\mathrm{EM}}(X_1) = y_1, \mathcal{M}_{\mathrm{EM}}(X_2) = y_2)} \Big/ \frac{\Pr(X_1 = x_1)}{\Pr(X_1 = x_2)} \right) \right|$$

$$= \left| \ln\left( \frac{\Pr(X_1 = x_1 \mid \mathcal{M}_{\mathrm{EM}}(X_1) = y_1, \mathcal{M}_{\mathrm{EM}}(X_2) = y_2)}{\Pr(X_1 = x_2 \mid \mathcal{M}_{\mathrm{EM}}(X_1) = y_1, \mathcal{M}_{\mathrm{EM}}(X_2) = y_2)} \right) \right|$$

$$= \left| \ln\left( \frac{\Pr(X_1 = x_1, \mathcal{M}_{\mathrm{EM}}(X_1) = y_1, \mathcal{M}_{\mathrm{EM}}(X_2) = y_2)}{\Pr(X_1 = x_2, \mathcal{M}_{\mathrm{EM}}(X_1) = y_1, \mathcal{M}_{\mathrm{EM}}(X_2) = y_2)} \right) \right|$$

$$= \left| \ln\left( \frac{\sum_{x \in \{x_1, x_2\}} \Pr(X_1 = x_1, X_2 = x, \mathcal{M}_{\mathrm{EM}}(X_1) = y_1, \mathcal{M}_{\mathrm{EM}}(X_2) = y_2)}{\sum_{x \in \{x_1, x_2\}} \Pr(X_1 = x_2, X_2 = x, \mathcal{M}_{\mathrm{EM}}(X_1) = y_1, \mathcal{M}_{\mathrm{EM}}(X_2) = y_2)} \right) \right|$$

$$= \left| \ln\left( \frac{\sum_{x \in \{x_1, x_2\}} \Pr(X_1 = x_1, X_2 = x) \Pr(\mathcal{M}_{\mathrm{EM}}(X_1) = y_1, \mathcal{M}_{\mathrm{EM}}(X_2) = y_2 \mid X_1 = x_1, X_2 = x)}{\sum_{x \in \{x_1, x_2\}} \Pr(X_1 = x_2, X_2 = x) \Pr(\mathcal{M}_{\mathrm{EM}}(X_1) = y_1, \mathcal{M}_{\mathrm{EM}}(X_2) = y_2 \mid X_1 = x_2, X_2 = x)} \right) \right|$$

$$= \left| \ln\left( \frac{\sum_{x \in \{x_1, x_2\}} \Pr(X_1 = x_1, X_2 = x) \Pr(\mathcal{M}_{\mathrm{EM}}(X_1) = y_1 \mid X_1 = x_1) \Pr(\mathcal{M}_{\mathrm{EM}}(X_2) = y_2 \mid X_2 = x)}{\sum_{x \in \{x_1, x_2\}} \Pr(X_1 = x_2, X_2 = x) \Pr(\mathcal{M}_{\mathrm{EM}}(X_1) = y_1 \mid X_1 = x_2) \Pr(\mathcal{M}_{\mathrm{EM}}(X_2) = y_2 \mid X_2 = x)} \right) \right|$$

$$= \left| \ln\left( \frac{0.01 \cdot 0.72 \cdot 0.28 + 0.49 \cdot 0.72 \cdot 0.72}{0.49 \cdot 0.28 \cdot 0.28 + 0.01 \cdot 0.28 \cdot 0.72} \right) \right|$$

$$= \left| \ln\left( \frac{0.256032}{0.040432} \right) \right|$$

$$= 1.8456 > \epsilon \tag{41}$$

## C.2   PER-USER ACCOUNTING AS A COMPLEMENTARY MITIGATION

Notably, when records can be cleanly grouped by user and the mechanism is explicitly designed with per-user accounting, a per-user privacy budget is a natural and effective way to mitigate composition across correlated records. In the classical DP setting, this corresponds to treating each user as the "unit of protection," ensuring that all contributions from the same user share a fixed budget and that repeated participation does not cause unbounded privacy loss.

However, *user grouping is not always known or reliable in the kinds of applications we target.* In many text- and embedding-based systems, the mechanism does not have (and arguably should not assume) a trusted user identifier for each record. Posts may come from multiple accounts controlled by the same person, or a single account may refer to several different individuals or secrets. Determining that a set of words, embeddings, or snippets describe the same person or the same underlying secret is itself part of the adversarial inference task, for example, linking posts across accounts, linking mentions of the same individual across different documents, or reconstructing relationships such as family ties. In such settings, a per-user budget implicitly assumes that this partition into users is known and enforced by the defender, whereas our threat model explicitly allows the adversary to aggregate any correlated releases they can link.

This issue is also reflected in our case study. The public datasets we use do not contain user identifiers or reliable group labels that could serve as a ground truth for "per-user" segmentation. Constructing a per-user baseline would therefore require introducing additional, task-specific grouping heuristics (e.g., clustering by metadata, text similarity, or co-occurrence), which are orthogonal to our core threat model and may either underestimate or overestimate correlation. Since our goal is to study joint leakage over arbitrary correlated secrets, without assuming that the defender knows how records should be grouped, we deliberately adopt a user-agnostic formulation of mPL and AmPL.

We see per-user budgeting as a complementary mitigation, not as an alternative to our framework. In applications where reliable user identifiers and grouping assumptions are available, our mechanisms and audits can be combined with per-user accounting: the system can enforce a per-user privacy budget while our framework still evaluates whether correlated secrets within or across those groups violate the intended metric privacy guarantees. We will clarify this perspective in the revised paper and explicitly discuss per-user accounting as a practical extension in deployments that have trustworthy user IDs.

## C.3 GENERALITY BEYOND TEXT EMBEDDINGS.

Although our experiments focus on textual embeddings, both the joint-leakage notion (mPL) and the AmPL repair framework are, by construction, modality-agnostic. The mPL definition only assumes (i) a metric $d$ over the space of secrets and (ii) a task-specific utility loss, while AmPL requires access to (iii) a representation space in which the mechanism operates and (iv) a learned attacker that estimates posteriors from perturbed outputs. None of these ingredients are specific to text. In particular, analogous joint-leakage risks arise whenever multiple correlated releases about the same underlying secret are produced in other modalities: for example, frames or crops of the same image (vision), longitudinal records for the same entity (tabular or time-series data), or multiple recordings of the same speaker or event (audio). In each case, an adversary can train a multi-input model to aggregate correlated observations and potentially violate per-release privacy guarantees, exactly as we demonstrate in the textual setting.

We choose text embeddings as our primary case study because (i) they are widely deployed in high-stakes applications (e.g., LLM-based interfaces, retrieval-augmented systems, and embedding-based search), (ii) the threat of cross-record aggregation is especially acute due to rich semantic correlations, and (iii) covering multiple modalities at similar depth would significantly expand the scope of the paper.

## C.4 PERTURBATION SPACE: EMBEDDINGS VS. TEXT.

Our current implementation instantiates AmPL by perturbing word embeddings, because embeddings are a common interface in modern NLP systems (e.g., for search, retrieval, and recommendation). However, the *framework* itself is agnostic to whether perturbations are applied in embedding space or directly on text.

Formally, let $\mathcal{X}$ denote the space of original text (words or sentences) and $\mathcal{Y}$ the space of perturbed outputs (which may be text or embeddings). Any defense mechanism that specifies a randomized map

$$M : \mathcal{X} \to \mathsf{Distr}(\mathcal{Y}), \qquad x \mapsto \mathcal{M}(\cdot \mid x)$$

induces a perturbation channel and a corresponding posterior $\Pr[X \mid Y]$. Our mPL definition and learned-adversary attack are defined purely in terms of this posterior, and therefore apply unchanged to *any* such stochastic channel.

In particular, defenses that act directly on text, such as token deletion, insertion of noise characters, or synonym substitution, still define a stochastic mapping from original text $x$ to perturbed text $y$. An attacker observing $y$ can then process it through the same embedding model (or any other feature extractor) and train a predictor exactly as in our experiments. From the perspective of mPL and the learned adversary, the only requirement is that $(X, Y)$ are jointly distributed via some randomized mechanism; the choice of operating in embedding space or text space is an implementation detail of the defense, not a limitation of the framework.

# D OMITTED PROOFS

## D.1 PROOF OF PROPOSITION 1 (POST-PROCESSING FOR BOUNDED MPL)

**Proposition 1.** *(**Post-processing for bounded mPL**) Suppose the data perturbation mechanism $\mathcal{M}$ satisfies the $\epsilon$-bounded joint mPL constraint: $\sup_{x_i \neq x_j} \sup_{\mathbf{y} \in \mathcal{Y}^L} \mathrm{mPL}_{\mathcal{M}}(x_i, x_j, \mathbf{y}) \leq \epsilon$. Then for any function $f$, the post-processed output $f \circ \mathcal{M}$ also satisfies the bounded joint mPL constraint:*

$$\sup_{x_i \neq x_j} \sup_{\mathbf{y} \in \mathcal{Y}^L} \mathrm{mPL}_{f \circ \mathcal{M}}(x_i, x_j, \mathbf{z}) \leq \epsilon. \tag{42}$$

*where $\mathcal{Z} = \mathrm{Range}(f \circ \mathcal{M})$..*

*Proof.* Let $f^{-1}(\mathbf{z}) = \{\mathbf{y} : f(\mathbf{y}) = \mathbf{z}\}$ denote the preimage of $z$. Fix any pair $x_i, x_j \in \mathcal{X}$ and any $\mathbf{y} \in \mathcal{Y}^L$, we have

$$\text{mPL}_{f \circ \mathcal{M}}(x_i, x_j, \mathbf{y}) \tag{43}$$

$$= \ln \left( \frac{\Pr(X_\ell = x_i | \{f \circ \mathcal{M}(X_1), \ldots, f \circ \mathcal{M}(X_L)\} = \mathbf{z})}{\Pr(X_\ell = x_j | \{f \circ \mathcal{M}(X_1), \ldots, f \circ \mathcal{M}(X_L)\} = \mathbf{z})} \middle/ \frac{\Pr(X_\ell = x_i)}{\Pr(X_\ell = x_j)} \right) \tag{44}$$

$$= \ln \left( \frac{\Pr(X_\ell = x_i, \{\mathcal{M}(X_1), \ldots, \mathcal{M}(X_L)\} = f^{-1}(\mathbf{z}))}{\Pr(X_\ell = x_j, \{\mathcal{M}(X_1), \ldots, \mathcal{M}(X_L)\} = f^{-1}(\mathbf{z}))} \middle/ \frac{\Pr(X_\ell = x_i)}{\Pr(X_\ell = x_j)} \right) \tag{45}$$

$$\leq d_{x_i, x_j} \epsilon. \tag{46}$$

$\square$

## D.2 Proof of Proposition 2 (Single-Observation Equivalence of mPL and mDP)

**Proposition 2** (Single-observation equivalence of mPL and mDP). *Let $\mathcal{M}$ be a perturbation mechanism on a metric secret space $(\mathcal{X}, d)$. Define the single-observation mPL for a pair $(x_i, x_j)$ and observation $y$ by*

$$\text{mPL}_{\mathcal{M}}((x_i, x_j), y) = \frac{1}{d_{x_i, x_j}} \left| \ln \frac{\Pr(X = x_i \mid \mathcal{M}(X) = y)}{\Pr(X = x_j \mid \mathcal{M}(X) = y)} - \ln \frac{\Pr(X = x_i)}{\Pr(X = x_j)} \right|. \tag{47}$$

*For any $\epsilon \geq 0$, $\mathcal{M}$ satisfies $(\epsilon, d)$-mDP if and only if the single-observation mPL bound holds, i.e., $\sup_{x_i \neq x_j} \sup_{y \in \mathcal{Y}} \text{mPL}_{\mathcal{M}}((x_i, x_j), y) \leq \epsilon$.*

*Proof.* Fix any pair $\Pr(X = x_i), \Pr(X = x_j) > 0$. By Bayes' rule,

$$\frac{\Pr[X = x_i \mid \mathcal{M}(X) = y]}{\Pr[X = x_j \mid \mathcal{M}(X) = y]} = \frac{\Pr[\mathcal{M}(X) = y \mid X = x_i]}{\Pr[\mathcal{M}(X) = y \mid X = x_j]} \cdot \frac{\Pr(X = x_i)}{\Pr(X = x_j)}. \tag{48}$$

$$\Leftrightarrow \ln \frac{\Pr[X = x_i \mid \mathcal{M}(X) = y]}{\Pr[X = x_j \mid \mathcal{M}(X) = y]} - \ln \frac{\Pr(X = x_i)}{\Pr(X = x_j)} = \ln \frac{\Pr[\mathcal{M}(X) = y \mid X = x_i]}{\Pr[\mathcal{M}(X) = y \mid X = x_j]}. \tag{49}$$

Therefore

$$\underbrace{\frac{1}{d_{x_i, x_j}} \left| \ln \frac{\Pr(X = x_i \mid \mathcal{M}(X) = y)}{\Pr(X = x_j \mid \mathcal{M}(X) = y)} - \ln \frac{\Pr(X = x_i)}{\Pr(X = x_j)} \right| \leq \epsilon}_{\text{Pointwise form of bounded mPL constraint}} \tag{50}$$

$$\Leftrightarrow \underbrace{\ln \frac{\Pr[\mathcal{M}(X) = y \mid X = x_i]}{\Pr[\mathcal{M}(X) = y \mid X = x_j]} \leq \epsilon \, d_{x_i, x_j}}_{\text{Pointwise form of mDP}}. \tag{51}$$

which concludes the proof. $\square$

## D.3 Proof of Proposition 3 (Independent-Observation Equivalence of mPL and mDP)

**Proposition 3** (Independent-observation equivalence of mPL and mDP). *If the $L$ secret words $X_1, \ldots, X_L$ are independently distributed, then ensuring $\sup_{x_i \neq x_j} \sup_{y^\ell \in \mathcal{Y}} \text{mPL}_{\mathcal{M}}((x_i, x_j), y^\ell) \leq \epsilon$ for each $y^\ell$ ($\ell = 1, ..., L$) is sufficient to guarantee $\sup_{x_i \neq x_j} \sup_{\mathbf{y} \in \mathcal{Y}^L} \text{mPL}_{\mathcal{M}}(x_i, x_j, \mathbf{y}) \leq \epsilon$.*

*Proof.* First, if the random variables $(X_\ell, Z_l)$ are independent with $(X_t, Z_t)$, and $\tilde{\mathcal{M}}$ is a mearable function, then $\tilde{\mathcal{M}}(X_\ell, Z_l)$ and $\tilde{\mathcal{M}}(X_t, Z_t)$ are independent, as measurable functions preserve independence in probability theory [reference].

Then, we can obtain

$$\Pr\left[X_\ell = x \mid \left\{\tilde{\mathcal{M}}(X_1, Z), \ldots, \tilde{\mathcal{M}}(X_L, Z)\right\} = \mathbf{y}\right] \tag{52}$$

$$= \frac{\Pr\left[X_\ell = x, \left\{\tilde{\mathcal{M}}(X_1, Z), \ldots, \tilde{\mathcal{M}}(X_L, Z)\right\} = \mathbf{y}\right]}{\Pr\left[\left\{\tilde{\mathcal{M}}(X_1, Z), \ldots, \tilde{\mathcal{M}}(X_L, Z)\right\} = \mathbf{y}\right]} \tag{53}$$

$$= \frac{\prod_{t=1, t\neq\ell}^{L} \Pr\left[\tilde{\mathcal{M}}(X_t, Z) = y_t\right] \Pr\left[X_\ell = x, \tilde{\mathcal{M}}(X_\ell, Z) = y_\ell\right]}{\prod_{t=1}^{L} \Pr\left[\tilde{\mathcal{M}}(X_t, Z) = y_t\right]} \tag{54}$$

$$= \Pr\left[X_\ell = x \mid \tilde{\mathcal{M}}(X_\ell, Z) = y_\ell\right] \tag{55}$$

$$\square$$

### D.4 PROOF OF PROPOSITION 4 (CONCENTRATION GUARANTEES)

**Proposition 4.** *[Concentration Guarantee for Probabilistic mDP Sampling] If the empirical violation rate satisfies $\hat{p}_{\mathcal{S}_\ell} = \xi\delta$ for some constant $\xi < 1$, then $\Pr\left[p_{\mathcal{X}_\ell^2} \leq \delta\right] \geq 1 - 2\exp(-2S_\ell(1-\xi)^2\delta^2)$.*

*Proof.* Suppose the empirical violation rate satisfies $\hat{p}_{\mathcal{S}} = \xi\delta$ for some constant $\xi < 1$. Our goal is to show that, with high probability, the true violation probability $p_{\mathcal{X}^2}$ is at most $\delta$.

First, by Hoeffding's inequality, for any $t > 0$:

$$\Pr\left[|\hat{p}_{\mathcal{S}} - p_{\mathcal{X}^2}| > t\right] \leq 2e^{-2S_\ell t^2}. \tag{56}$$

Let $t = (1 - \xi)\delta$. Then:

$$\Pr\left[|\hat{p}_{\mathcal{S}} - p_{\mathcal{X}^2}| > (1-\xi)\delta\right] \leq 2e^{-2S_\ell(1-\xi)^2\delta^2}. \tag{57}$$

This implies:

$$\Pr\left[|\hat{p}_{\mathcal{S}} - p_{\mathcal{X}^2}| \leq (1-\xi)\delta\right] \geq 1 - 2e^{-2S_\ell(1-\xi)^2\delta^2}. \tag{58}$$

Now, under the event that $|\hat{p}_{\mathcal{S}} - p_{\mathcal{X}^2}| \leq (1-\xi)\delta$, we have:

$$p_{\mathcal{X}^2} \leq \hat{p}_{\mathcal{S}} + (1-\xi)\delta = \xi\delta + (1-\xi)\delta = \delta. \tag{59}$$

Thus:

$$\Pr\left[p_{\mathcal{X}^2} \leq \delta\right] \geq 1 - 2e^{-2S_\ell(1-\xi)^2\delta^2}, \tag{60}$$

which completes the proof. $\square$

### D.5 PROOF OF PROPOSITION 5 (ASYMPTOTIC FAITHFULNESS OF THE mPL AUDIT)

**Proposition 5** (Asymptotic faithfulness of the mPL audit). *Let $p(x \mid y)$ denote the true posterior and $q_\theta(x \mid y)$ the adversary trained on $n$ i.i.d. pairs $(x_i, y_i)$ by minimizing empirical conditional cross-entropy over a fixed neural-network class. Assume:*

   *A1. (Clipped posteriors) $\mathcal{X}$ is finite, and there exists $\gamma > 0$ such that $p(x \mid y) \geq \gamma$ and $q_\theta(x \mid y) \geq \gamma$ for all $x \in \mathcal{X}$ and all $y$ (implemented in practice by softmax clipping).*

   *A2. (Metric regularity) The metric satisfies $d(x_i, x_j) \geq d_{\min} > 0$ whenever $x_i \neq x_j$.*

*Then for any fixed pair of candidates $x_i, x_j$ there exists a constant $C > 0$ (depending only on $\gamma$, $d_{\min}$, and the candidate set) such that*

$$\mathbb{E}_Y\left[\left|\mathrm{mPL}(x_i, x_j; Y) - \widetilde{\mathrm{mPL}}(x_i, x_j; Y)\right|\right] \leq C\, n^{-\alpha/2} \tag{61}$$

*for all sufficiently large $n$, where $\mathrm{mPL}$ and $\widetilde{\mathrm{mPL}}$ denote the mPL computed using $p$ and $q_\theta$, respectively.*

**Lemma 1** (Cross-entropy and expected KL). *Define the population cross-entropy loss*

$$\mathcal{L}(\theta) := \mathbb{E}_{(X,Y)}\big[-\log q_\theta(X \mid Y)\big]. \tag{62}$$

*Then*

$$\mathcal{L}(\theta) = \mathbb{E}_Y\big[H\big(p(\cdot \mid Y)\big)\big] + \mathbb{E}_Y\big[\mathrm{KL}\big(p(\cdot \mid Y) \,\|\, q_\theta(\cdot \mid Y)\big)\big], \tag{63}$$

*where $H\big(p(\cdot \mid Y)\big)$ is the conditional entropy of the true posterior, therefore minimizing $\mathcal{L}(\theta)$ is equivalent (up to an additive constant) to minimizing the expected posterior KL divergence $\mathbb{E}_Y[\mathrm{KL}(p(\cdot \mid Y) \,\|\, q_\theta(\cdot \mid Y))]$.*

*Proof.* For any fixed $y$,

$$\mathbb{E}_{X|Y=y}\big[-\log q_\theta(X \mid y)\big] = \sum_x p(x \mid y)\big(-\log q_\theta(x \mid y)\big). \tag{64}$$

Add and subtract $\log p(x \mid y)$:

$$\sum_x p(x \mid y)\big(-\log q_\theta(x \mid y)\big) = \sum_x p(x \mid y)\big(-\log p(x \mid y)\big) + \sum_x p(x \mid y) \log \frac{p(x \mid y)}{q_\theta(x \mid y)}$$

$$= H\big(p(\cdot \mid y)\big) + \mathrm{KL}\big(p(\cdot \mid y) \,\|\, q_\theta(\cdot \mid y)\big).$$

Taking expectation over $Y$ yields the claim:

$$\mathcal{L}(\theta) = \mathbb{E}_Y\big[H\big(p(\cdot \mid Y)\big)\big] + \mathbb{E}_Y\big[\mathrm{KL}\big(p(\cdot \mid Y) \,\|\, q_\theta(\cdot \mid Y)\big)\big]. \tag{65}$$

$\square$

**Lemma 2** (Lipschitz stability of mPL w.r.t. posterior). *Suppose Assumptions A1 and A2 hold. Then for any $x_i, x_j, y$,*

$$\big|\mathrm{mPL}(x_i, x_j; y) - \widetilde{\mathrm{mPL}}(x_i, x_j; y)\big| \leq \frac{2}{\gamma d_{\min}} \big\|p(\cdot \mid y) - q_\theta(\cdot \mid y)\big\|_1. \tag{66}$$

*Proof.* By definition, the posterior-dependent part of mPL is a log-odds term of the form

$$g(a, b) := \log \frac{a}{b}, \tag{67}$$

where $a = p(x_i \mid y)$ and $b = p(x_j \mid y)$ for $\mathrm{mPL}$, and $a' = q_\theta(x_i \mid y)$, $b' = q_\theta(x_j \mid y)$ for $\widetilde{\mathrm{mPL}}$.

On the domain $[\gamma, 1 - \gamma]^2$, the gradient is

$$\nabla g(a, b) = \left(\frac{1}{a}, -\frac{1}{b}\right), \tag{68}$$

and by Assumption **A1**

$$\left|\frac{1}{a}\right| \leq \frac{1}{\gamma}, \qquad \left|\frac{1}{b}\right| \leq \frac{1}{\gamma}. \tag{69}$$

Thus the $\ell_1$-norm of the gradient is bounded:

$$\|\nabla g(a, b)\|_1 = \left|\frac{1}{a}\right| + \left|\frac{1}{b}\right| \leq \frac{2}{\gamma}. \tag{70}$$

By the mean value theorem, this implies a Lipschitz bound

$$\big|g(a, b) - g(a', b')\big| \leq \frac{2}{\gamma} \|(a, b) - (a', b')\|_1. \tag{71}$$

Applying this with

$$a = p(x_i \mid y), \quad b = p(x_j \mid y), \quad a' = q_\theta(x_i \mid y), \quad b' = q_\theta(x_j \mid y), \tag{72}$$

and using the lower bound on the metric distance $d(x_i, x_j) \geq d_{\min} > 0$ from Assumption **A2**, we obtain

$$\big|\mathrm{mPL}(x_i, x_j; y) - \widetilde{\mathrm{mPL}}(x_i, x_j; y)\big| \leq \frac{2}{\gamma d_{\min}} \big\|(a, b) - (a', b')\big\|_1. \tag{73}$$

Finally, since $(a, b)$ and $(a', b')$ are sub-vectors of $p(\cdot \mid y)$ and $q_\theta(\cdot \mid y)$, we have

$$\big\|(a, b) - (a', b')\big\|_1 \leq \big\|p(\cdot \mid y) - q_\theta(\cdot \mid y)\big\|_1, \tag{74}$$

which gives the claimed bound. $\square$

*Proof of Proposition 5.* In practice, we train $q_\theta$ by minimizing the empirical conditional cross-entropy

$$\hat{\mathcal{L}}(\theta) = \frac{1}{n} \sum_{i=1}^{n} -\log q_\theta(x_i \mid y_i), \tag{75}$$

which is a Monte Carlo estimate of the population loss

$$\mathcal{L}(\theta) = \mathbb{E}_{(X,Y)}\big[-\log q_\theta(X \mid Y)\big]. \tag{76}$$

Generalization bounds for deep networks trained with cross-entropy, based on spectral norms and margins, show that with high probability over the draw of the training sample, the excess population cross-entropy of the trained model over the best-in-class predictor decays at a polynomial rate in $n$ (see, e.g., Bartlett et al. (2017)). Combined with Lemma 1, this implies that there exist constants $C_0, \alpha > 0$ such that, for sufficiently large $n$,

$$\mathbb{E}_Y\big[\mathrm{KL}\big(p(\cdot \mid Y) \,\|\, q_\theta(\cdot \mid Y)\big)\big] \ \leq\ C_0 \, n^{-\alpha}. \tag{77}$$

Next, by Lemma 2 and Pinsker's inequality,

$$\big\|p(\cdot \mid y) - q_\theta(\cdot \mid y)\big\|_1 \ \leq\ \sqrt{2\,\mathrm{KL}\big(p(\cdot \mid y) \,\|\, q_\theta(\cdot \mid y)\big)}, \tag{78}$$

we obtain, for each $y$,

$$\big|\mathrm{mPL}(x_i, x_j; y) - \widetilde{\mathrm{mPL}}(x_i, x_j; y)\big| \ \leq\ \frac{2\sqrt{2}}{\gamma d_{\min}} \sqrt{\mathrm{KL}\big(p(\cdot \mid y) \,\|\, q_\theta(\cdot \mid y)\big)}. \tag{79}$$

Taking expectation over $Y$ and applying Jensen's inequality to the concave function $z \mapsto \sqrt{z}$ yields

$$\mathbb{E}_Y\Big[\big|\mathrm{mPL}(x_i, x_j; Y) - \widetilde{\mathrm{mPL}}(x_i, x_j; Y)\big|\Big] \ \leq\ \frac{2\sqrt{2}}{\gamma d_{\min}} \sqrt{\mathbb{E}_Y\Big[\mathrm{KL}\big(p(\cdot \mid Y) \,\|\, q_\theta(\cdot \mid Y)\big)\Big]}. \tag{80}$$

Using the generalization bound $\mathbb{E}_Y[\mathrm{KL}(p(\cdot \mid Y) \,\|\, q_\theta(\cdot \mid Y))] \leq C_0 n^{-\alpha}$ then gives

$$\mathbb{E}_Y\Big[\big|\mathrm{mPL}(x_i, x_j; Y) - \widetilde{\mathrm{mPL}}(x_i, x_j; Y)\big|\Big] \ \leq\ \frac{2\sqrt{2C_0}}{\gamma d_{\min}} \, n^{-\alpha/2}. \tag{81}$$

Setting $C = 2\sqrt{2C_0}/(\gamma d_{\min})$ proves (61) and the proposition. $\qquad\square$

# E    ADDITIONAL DETAILS OF CASE STUDY

In this section, we provide additional details for the case study—including the design of the data perturbation and utiltiy loss—serving as complementary material to Section 4 of the main paper.

## E.1    2-LEVEL DATA PERTURBATION

Let $\mathcal{U}$ denote the complete set of word embeddings in the dataset. We define $\mathcal{X}_1$ and $\mathcal{X}_2$ as the subsets corresponding to PII and PoII embeddings, respectively. The overall set of sensitive embeddings is given by $\mathcal{X} = \mathcal{X}_1 \cup \mathcal{X}_2$, which is a subset of the full embedding set, i.e., $\mathcal{X} \subseteq \mathcal{U}$. To identify clear instances of PII, we apply *Named Entity Recognition (NER)* using the spaCy Python library[1]. Specifically, we assign a Level 1 privacy label to any word token identified by the NER model as a likely PII entity—such as PERSON, GPE, or ORG. These Level 1 tokens form the subset $\mathcal{X}_1 \subset \mathcal{U}$.

For the complementary set $\mathcal{U} \setminus \mathcal{X}_1$, we adopt the cosine-similarity-based approach proposed by Hassan et al. Hassan et al. (2023) to identify *Potentially Identifiable Information (PoII)*. Specifically, we compute the cosine similarity between each token in $\mathcal{U} \setminus \mathcal{X}_1$ and the tokens in $\mathcal{X}_1$ using pretrained GloVe embeddings. The top 10% of tokens from $\mathcal{U} \setminus \mathcal{X}_1$ with the highest similarity scores are labeled as Level 2 privacy and form the PoII set $\mathcal{X}_2$. All remaining tokens—those not classified as either PII or PoII—are treated as non-sensitive.

---

[1]spaCy: Industrial-Strength Natural Language Processing in Python. Available at: https://spacy.io

## E.2 UTILITY LOSS CALCULATION

The *utility loss* quantifies the impact of the obfuscation process on sentence quality. For a sensitive token embedding $x_i$ replaced by a perturbed embedding $y_k$, the loss is defined as

$$c_{i,k} = 1 - \frac{Sim(x_i, y_k) + 1}{2}, \tag{82}$$

where $Sim(x_i, y_k)$ denotes the cosine similarity between $x_i$ and $y_k$: $Sim(x_i, y_k) = \frac{x_i \cdot y_k}{\|x_i\| \, \|y_k\|}$. The normalization term $\frac{(\cdot)+1}{2}$ maps the cosine similarity from the range $[-1, 1]$ to $[0, 1]$, ensuring that $c_{i,k} \in [0, 1]$. Thus, $c_{i,k}$ captures the semantic deviation introduced by the perturbation, with higher values indicating greater semantic loss. Each token embedding is represented using pre-trained 100-dimensional GloVe vectors, which preserve the structure and context of the original sentence. The overall utility loss for $(x_i, y_k)$ is computed over all sensitive tokens and candidate replacements, ensuring that the semantic structure is preserved as faithfully as possible.

The experiment is performed for multiple values of $\epsilon$, the resulting utility loss stores the variations corresponding to the varying privacy guarantees on the semantic utility. The final matrix provides the insights about the trade-offs between privacy preservation and utility.

# F ADDITIONAL EXPERIMENTAL RESULTS

## F.1 RECOMMENDED $\delta$ THRESHOLDS AND FAILURE BOUND ANALYSIS

Table 3: Estimated achievable threshold $\tilde{\delta} = 1.05 \, \delta^\star$ (5% margin) and the range of PBmPL failure bounds across adversaries, reported as $\log_{10} p_{\text{fail}}$.

| Dataset ($\epsilon$) | $\tilde{\delta}$ | Range of $\log_{10} p_{\text{fail}}$ |
|---|---|---|
| AG News ($\epsilon = 0.3$) | 0.063000 | $[-7967, -87]$ |
| AG News ($\epsilon = 0.4$) | 0.025620 | $[-1969, -14]$ |
| AG News ($\epsilon = 0.5$) | 0.010080 | $[-488, -2]$ |
| IMDB ($\epsilon = 0.3$) | 0.136500 | $[-790639, -1803]$ |
| IMDB ($\epsilon = 0.4$) | 0.054915 | $[-128719, -292]$ |
| IMDB ($\epsilon = 0.5$) | 0.020895 | $[-18635, -42]$ |
| Amazon ($\epsilon = 0.3$) | 0.108255 | $[-86243, -761]$ |
| Amazon ($\epsilon = 0.4$) | 0.051765 | $[-25734, -174]$ |
| Amazon ($\epsilon = 0.5$) | 0.024150 | $[-5984, -38]$ |

To ensure that the probabilistic bound on posterior leakage (PBmPL) is both valid and practically meaningful, we calibrate $\delta$ to the maximum observed violation ratio for each dataset and $\epsilon$, with a 5% margin to guarantee $\xi < 1$ across all adversaries. Table 3 reports the recommended thresholds $\tilde{\delta} = 1.05 \, \delta^\star$ along with the corresponding ranges of $\log_{10} p_{\text{fail}}$. The results show that even under these tighter settings, PBmPL failure probabilities remain astronomically small, confirming that violations are virtually impossible at scale.

## F.2 EXPECTED UTILITY LOSS WHEN APPLYING RNN AND LSTM AS ADVERSARIAL MODELS

Figures 5–7 report *expected utility loss* for the same method set and ($\epsilon \in \{0.30, 0.40, 0.50\}$) configuration, differing only by the adversary: Fig. 4 uses a *Transformer*, Fig. 5 an *RNN*, and Fig. 6 an *LSTM*; in each figure, panels (a)–(c) correspond to AG News, IMDB, and Amazon, and bars are grouped by methods (EM/AmPL variants, with and without the remapping step, RMP). Across all datasets and adversaries, utility loss *monotonically decreases* as $\epsilon$ increases (privacy–utility trade-off); within the AmPL family, AMPL-U consistently achieves the *lowest* or near-lowest loss (by design), while AMPL closely follows with a small gap and AMPL-1/AMPL-P typically lie between AMPL and EM. Enabling RMP further *reduces* loss compared to the corresponding "no RMP" variants. The *relative ordering* of methods is stable across Fig. 4 (Transformer), Fig. 5 (RNN), and Fig. 6 (LSTM),

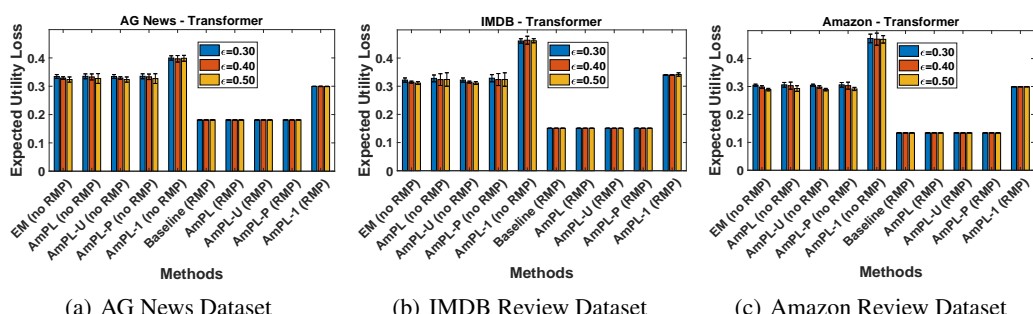

(a) AG News Dataset  (b) IMDB Review Dataset  (c) Amazon Review Dataset

Figure 5: Expected utility loss (applying Transformer as the adversarial model).

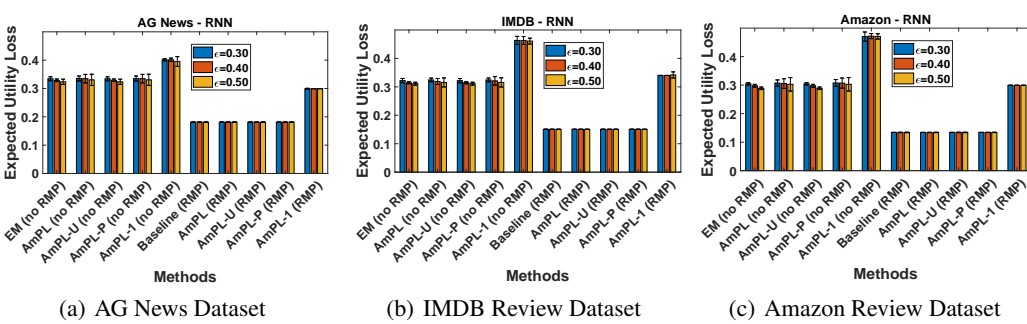

(a) AG News Dataset  (b) IMDB Review Dataset  (c) Amazon Review Dataset

Figure 6: Expected utility loss (applying RNN as the adversarial model).

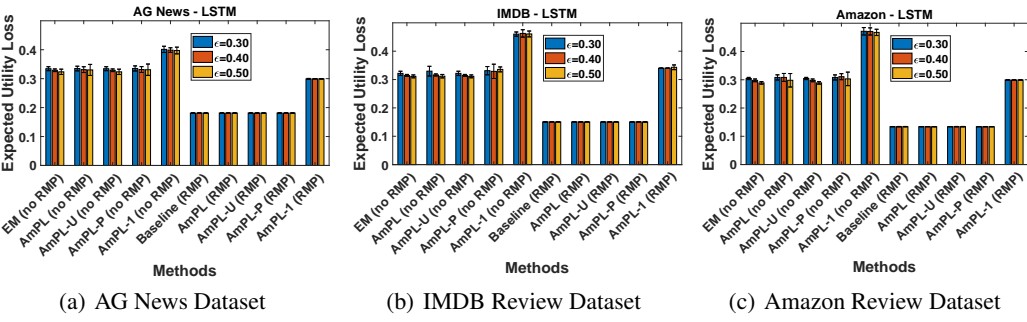

(a) AG News Dataset  (b) IMDB Review Dataset  (c) Amazon Review Dataset

Figure 7: Expected utility loss (applying LSTM as the adversarial model).

indicating robustness to the attacker model; dataset-wise, Amazon shows the largest absolute losses, followed by AG News and IMDB, but the method ranking and $\epsilon$-sensitivity remain consistent.

We also observe that, across all datasets, adversaries, and $\epsilon \in \{0.30, 0.40, 0.50\}$, enabling RMP yields a large and consistent reduction in expected utility loss. For the core methods (EM/AmPL/AmPL-U/AmPL-P), RMP lowers loss from $\approx 0.33$ (no-RMP) to a fixed $\approx 0.18$ on *AG News* (about *46%* reduction), from $\approx 0.32$ to $\approx 0.15$ on *IMDB* (about *53%*), and from $\approx 0.30$ to $\approx 0.134$ on *Amazon* (about *56%*); the exact percentages are stable across adversaries (Transformer/RNN/LSTM) and $\epsilon$. For *AmPL-1*, RMP still helps but less dramatically, improving from $\approx 0.40 \rightarrow 0.299$ on AG News (*25%*), $\approx 0.46 \rightarrow 0.340$ on IMDB (*26–27%*), and $\approx 0.47 \rightarrow 0.299$ on Amazon (*36%*). Overall, RMP delivers a ∼*45–56%* reduction for EM/AmPL-family methods and a *25–36%* reduction for AmPL-1, effectively halving the utility loss in most settings.

F.3 TRADEOFF BETWEEN UTILITY AND VIOLATION RATE

Figure 8 illustrates the trade-off between utility loss and the empirical mPL violation ratio for AmPL without Bayesian remap. Each point corresponds to one configuration of the mechanism, obtained by varying $\alpha_1, \alpha_2 \in [0.1, 1.0]$ with step $0.1$, evaluated at base privacy level $\varepsilon = 0.5$ on $11{,}214{,}777$ $(x_i, x_j, y)$ triples. The scatter plot shows that as the violation ratio decreases (moving left), the utility loss generally increases, illustrating the expected privacy–utility trade-off: configurations that inject less noise achieve lower utility loss but suffer higher violation ratios, whereas configurations enforcing lower violation ratios incur slightly higher utility loss.

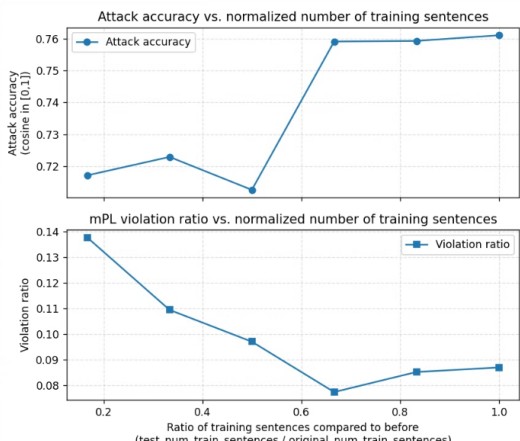

Figure 8: Trade-off between empirical mPL violation ratio and utility loss for AmPL without Bayesian remap under a Transformer-based attacker.

### F.4 EFFECT OF ATTACKER TRAINING DATA SIZE

To assess how many supervised pairs the learned adversary requires, we run a learning-curve experiment on AGNEWS. We subsample the adversary's training set to different fractions $r \in (0, 1]$ of the original size, retrain the attack for each $r$, and report (i) the attack accuracy, measured as the average cosine similarity between the reconstructed and ground-truth embeddings, and (ii) the fraction of $(x_i, x_j, y)$ triples that violate the mPL threshold. Figure 9 shows that the attack saturates quickly: using only about 60% of the supervised training pairs already attains nearly the same cosine accuracy and mPL violation ratio as training on the full dataset.

Interestingly, the estimated violation ratio *slightly decreases* as the number of training pairs increases. Recall that our posterior-leakage metric is defined in terms of how much the posterior odds ratio $\frac{q_\theta(x_i|y)}{q_\theta(x_j|y)}$ deviates from the prior odds $\frac{p(x_i)}{p(x_j)}$. With more training data, the two attack models are better calibrated and their posterior ratios move closer to the true posterior odds, so the estimated posterior leakage converges to its true value. With limited data, the learned scores are noisier and can become over-confident on some triples $(x_i, x_j, y)$, which can artificially inflate the deviation between posterior and prior odds and hence yield a higher apparent violation ratio. Thus, increasing the training size yields more accurate (and sometimes smaller) estimates of posterior leakage, which explains the downward trend in the violation curve.

Figure 9: **Effect of attacker training data size.** Top: attack accuracy (cosine similarity) as a function of the normalized number of training sentences. Bottom: mPL violation ratio as a function of the normalized number of training sentences.

## G RELATED WORK

Since the introduction of metric differential privacy (mDP) by Chatzikokolakis et al. Chatzikokolakis et al. (2013), substantial effort has been devoted to designing perturbation mechanisms that satisfy mDP while preserving downstream utility. Existing methods can be broadly grouped into two families: *predefined noise distribution mechanisms*, which sample outputs from a fixed distribution calibrated to the metric distance, and *optimization-based mechanisms*, which formulate mechanism design as an explicit utility-minimization problem under mDP constraints.

**Predefined noise distribution mechanisms.** The Laplace mechanism is a classical DP mechanism Dwork et al. (2006) that was adapted to mDP by Chatzikokolakis et al. Chatzikokolakis et al. (2013). Instead of scaling noise to a global sensitivity, the mDP variant calibrates the noise scale to the metric distance $d_\mathcal{X}$ between records, typically proportional to $d_\mathcal{X}/\epsilon$, so that outputs for nearby records remain statistically indistinguishable. This adaptation underlies Geo-Indistinguishability (Geo-Ind) Andrés et al. (2013) for location privacy: early work focused on sporadic location re-

leases Chatzikokolakis et al. (2013); Andrés et al. (2013), while subsequent extensions studied continuous tracking, real-time trajectories, and personalized budgets Chatzikokolakis et al. (2014); Hua et al. (2017); Yu et al. (2023); Min et al. (2023). Despite its simplicity and efficiency Carvalho et al. (2021), additive Laplace noise is poorly suited to structured domains (e.g., graphs or embeddings), where it may severely distort structure or ignore non-uniform density. For word embeddings, direct perturbation can produce invalid vectors, so mechanisms often project back to the nearest neighbor in the embedding space Fernandes et al. (2018); Feyisetan et al. (2019), implicitly assuming the embedding space is non-sensitive and potentially introducing additional privacy risks during neighbor retrieval.

Exponential mechanisms (EM) Carvalho et al. (2021); Imola et al. (2022) address some of these limitations by selecting outputs from discrete candidate sets according to a utility-driven quality score, thereby bypassing continuous additive noise. EM-based approaches have proved particularly effective in location privacy (trajectory protection Chatzikokolakis et al. (2015); Yu et al. (2017); spatial crowdsourcing Gursoy et al. (2019); Niu et al. (2020); Ren et al. (2022); Dong et al. (2019)), textual data (word substitution Wang et al. (2017b); Feyisetan et al. (2020b); embeddings Carvalho et al. (2021); Meisenbacher et al. (2024)), and graph data (edge-preserving graph perturbation Raskhodnikova & Smith (2016); private query responses Fioretto et al. (2019); Kamalaruban et al. (2020)). While EM offers greater flexibility and often better utility than the Laplace mechanism, it typically bases selection on the perturbation magnitude $d_{x,y}$ alone, which may not fully capture directional variations in utility loss across the output space. This limitation has motivated a line of *optimization-based* mechanisms that explicitly model both the magnitude and direction of utility changes for each candidate perturbation.

**Optimization-based mechanisms.** Optimization-based (often LP-based) mechanisms aim to minimize expected utility loss subject to mDP constraints. Since the seminal work of Bordenabe et al. Bordenabe et al. (2014), which formulated mechanism design under mDP as a linear program, a large body of work has explored LP-based constructions. These approaches typically represent mechanisms as stochastic matrices and optimize an LP objective that aggregates utility loss over all input–output pairs, while mDP constraints enforce bounded posterior changes for nearby inputs. Although such formulations are theoretically appealing, they require $O(|\mathcal{X}|^2)$ decision variables, rendering them computationally expensive for large domains. As a result, many implementations Bordenabe et al. (2014); Wang et al. (2016); Yu et al. (2017); Wang et al. (2017a) are restricted to relatively small domains with $|\mathcal{X}| \leq 100$. Bordenabe et al. Bordenabe et al. (2014) further proposed using spanners to approximate $d_{\mathcal{X}}$ via shortest-path distances in a sparse graph, but scalability remains limited; for example, solving the LP still takes on the order of 1,800 seconds when $|\mathcal{X}| = 400$ Imola et al. (2022).

To improve scalability, several works have introduced structural decompositions and hierarchical approximations. Ahuja et al. Ahuja et al. (2019) use a hierarchical index structure to protect location data under Geo-Ind. Decomposition-based methods, including Dantzig–Wolfe Qiu et al. (2020) and Benders decomposition Qiu (2024), partition large LPs into tractable subproblems that can be solved iteratively. Hybrid frameworks combining LP with probabilistic mechanisms have also emerged: Imola et al. Imola et al. (2022), for example, integrate a weighted EM with selective LP optimization on sensitive inputs, using local utility and sensitivity to adapt noise. Nevertheless, even state-of-the-art approaches typically handle domains of size at most $|\mathcal{X}| \leq 1,000$.

More recently, Qiu et al. Qiu et al. (2025) proposed a locality-aware optimization framework that restricts computation to spatial neighborhoods while aiming to preserve global privacy guarantees. This design significantly improves scalability, demonstrating applicability to domains with $|\mathcal{X}| = 1,600$, but at the cost of strict mDP compliance: the use of neighborhood-specific parameters can introduce subtle information leakage, which is problematic in applications requiring strong, global privacy guarantees.

**Connection to posterior leakage.** The mechanisms above are analyzed primarily through *a priori* mDP guarantees: they ensure that, for any pair of nearby inputs, the likelihood ratio of outputs is bounded by $e^\epsilon$. Posterior leakage–style measures instead evaluate privacy from the perspective of an explicit attacker, quantifying how much an adversary's posterior belief about a secret (or a set of correlated secrets) can be improved after observing the released outputs. In particular, posterior leakage captures (i) the *realized* inference power of concrete attack models, (ii) composition across multiple, potentially correlated releases, and (iii) violations arising from approximate or localized

implementations (e.g., spanner-based distances or neighborhood-restricted optimization) that still claim nominal mDP parameters. Our work is complementary to prior mechanism-design efforts: rather than proposing yet another LP or EM variant, we study how these mechanisms behave under posterior-leakage auditing and show that, despite satisfying or approximating mDP constraints on paper, they can induce significantly higher adversarial inference than their nominal $\epsilon$ would suggest.

