# OpenReview forum: "Metric-Normalized Posterior Leakage (mPL): Attacker-Aligned Privacy for Joint Consumption"
_ICLR.cc/2026/Conference — Submitted to ICLR 2026_

### Official Review · Reviewer_ynkJ · 2025-10-29

**Soundness:** 2
**Presentation:** 2
**Contribution:** 2
**Rating:** 4
**Confidence:** 4

**Summary:**

This paper identifies a significant gap in metric-style privacy for released embeddings: although metric differential privacy (mDP) protects each record in isolation, it can fail when an attacker jointly observes multiple, correlated releases. The authors formalize this joint posterior leakage (mPL) and propose an adaptive pipeline that uses learned adversaries to audit and adjust perturbation mechanisms.

**Strengths:**

The paper demonstrates that per-record mDP can be insufficient under *joint* consumption of multiple perturbed records. This is an important and well-motivated observation: attackers who aggregate correlated releases can substantially increase posterior confidence, undermining traditional mDP guarantees.

**Weaknesses:**

1. **Unspecified attacker capabilities (threat model).**
   - The threat model lacks a concrete specification of attacker knowledge and resources (e.g., access to candidate sets, ability to query models, prior distribution knowledge). A clearer, more explicit threat model is needed to interpret the empirical results and to understand when the reported mPL violations are realistic.

2. **Training data requirement for the learned adversary.**
   - How much paired data \((x, M(x))\) does the adversarial model require to reach the reported attack performance?
   - Practically, where would an attacker obtain sufficiently many *original* records and their corresponding perturbed releases to train such a model? If training requires a large amount of supervised pairs, the real-world applicability of the learned-adversary threat is reduced. The authors should report learning curves (attack accuracy / mPL violation vs. number of training pairs) and discuss plausible data-collection scenarios for the attacker.

3. **Assumed knowledge of the embedding method (line 388).**
   - The manuscript appears to assume the attacker knows the victim’s word-embedding method (L388). This is a strong assumption that strengthens the attacker considerably. Please clarify and justify this assumption: is it necessary for the attack to succeed, and how sensitive are results if the attacker uses a mismatched embedding model?

**Questions:**

1. **Generality beyond text.**
   - The experiments are limited to text embeddings. Do analogous joint-leakage risks arise for other modalities (images, tabular data, audio)? The paper should discuss whether the mPL phenomenon and the AmPL countermeasure generalize to non-text embeddings, or explicitly limit scope to textual embeddings.

2. **Perturbation model: embedding-level vs. token-level.**
   - The defense and attack operate on word embeddings (Exponential Mechanism over candidate embeddings). Would the learned-adversary attack still be effective if perturbations are applied directly on text (e.g., token deletion, insertion of noise characters, synonym substitution) rather than on embeddings? The paper should evaluate or at least discuss this alternative threat axis and its implications for attack success and practical defenses.

---

> ### Author Response · Authors · 2025-11-23
> **Response to Reviewr ynkJ**
>
> Thank you for providing constructive comments/suggestions. Below, we provide the response to the questions and comments.
>
> **Question 1: Generality beyond text**
>
> **Response:** Thank you for this valuable question. Our framework is not specific to text: the mPL definition only requires a metric over secrets and a utility loss, and AmPL only assumes access to an embedding space and a learned attacker, so in principle it applies to images, tabular data, audio, etc. In this paper, we focus on text embeddings because they are a widely used and high–stakes modality, and to keep the experimental scope manageable. In the revised paper, we have added the above clarification at the end of the **first paragraph of Section 4**, and a detailed discussion in **Appendix C.3**.
>
> We also evaluate mPL violation ratios on a tabular dataset to show that our method generalizes beyond text. Specifically, we use the *Breast Cancer Wisconsin (Diagnostic)* dataset, which contains 569 records with 30 continuous features each. All features are standardized (z-score) prior to perturbation. To reuse the same pipeline as in the text experiments, we treat each record as a single-token example, where the 30-dimensional standardized feature vector serves directly as the token embedding (i.e., as a vector input rather than text). In this non-text setting, every token is treated as sensitive (PII), and the PoII channel is empty. The tables below compare the violation ratios of AmPL and the EM baseline under different attack models, showing that AmPL continues to outperform EM when applied to protect non-text data, i.e., AmPL’s posterior leakage is **58.1%** lower than EM on average.
>
> **RNN (%)**
>
> | ε | 0.10 | 0.20 | 0.30 |
> |---|----|-----|----------|
> |EM (mDP)|35.40±23.42|19.56±20.59|14.23±12.45|
> |AmPL|13.42±19.44|3.66±4.34|0.98±2.44|
>
> **LSTM (%)**
>
> | ε | 0.10  | 0.20   | 0.30    |
> |---|------|--------|----|
> |EM (mDP) |14.06±27.64|14.18±23.37|12.67±13.70|
> | AmPL |4.74±4.37| 3.95±6.27| 2.71±3.58|
>
> **TRANSFORMER (%)**
>
> | ε | 0.10   | 0.20   | 0.30   |
> |----------|----|-----|---|
> | EM (mDP) | 48.49±23.78  | 29.88±25.96  | 19.49±12.88  |
> | AmPL     | 45.25±18.27  | 27.22±29.23  | 9.03±14.15   |
>
> **Question 2: Perturbation model: embedding-level vs. token-level**
>
> **Response:** Thank you for raising this point. While our current implementation perturbs word embeddings, the framework itself is agnostic to whether perturbations are applied in embedding space or directly to text. As long as there is a well-defined randomized map from an original word/sentence to a perturbed word/sentence, this induces a perturbation channel and a corresponding posterior, so our mPL definition and learned-adversary attack apply unchanged. In particular, a defense that performs token deletion, noise-character insertion, or synonym substitution still defines a stochastic mapping from original text to perturbed text; the attacker can then process the perturbed text through the same embedding model (or another feature extractor) and train a predictor exactly as we do. In the revised paper. we have added Section C.4 for a more detailed discussion.
>
> **Weakness 1: Unspecified attacker capabilities (threat model).**
>
> **Response:** We adopt standard security assumptions commonly used in the literature on privacy-preserving computing. The server is assumed to be *honest-but-curious*: it correctly follows the prescribed protocol but may attempt to infer sensitive information about individual users from the data it receives. We consider a prior-informed attacker whose goal is to identify the true secret $x_\ell$ from a candidate set $\mathcal{X}_\ell$ given one or more noisy releases. The attacker knows the perturbation mechanism $\mathcal{M}$ and its parameters, has access to an auxiliary corpus from the same population (to approximate the prior and train a posterior estimator), and can passively aggregate multiple correlated releases for the same secret. Given at least a single perturbed record $y$ and the mechanism $\mathcal{M}$, the server can infer the posterior distribution of $X$ using Bayes' rule. We have revised the paragraph of the **Adversarial model** section (immediately preceding Eq. (3)) to clarify our assumptions.

---

> ### Author Response · Authors · 2025-11-23
> **Response to Reviewr ynkJ (part 2)**
>
> **Weakness 2: Training data requirement for the learned adversary.**
>
> **Response:** We thank the reviewer for this insightful comment. First, we would like to clarify that our threat model intentionally assumes a strong attacker; from a defender’s perspective this is conservative rather than problematic, as we want to reason about worst-case leakage. Such an attacker does not need pre-existing supervised pairs from production logs: given white-box knowledge of the mechanism and access to an auxiliary corpus of candidate secrets (e.g., public text or embeddings), they can query the mechanism to generate arbitrarily many synthetic training pairs $(x, \mathcal{M}(x))$, exactly as in standard DP analyses where the adversary is allowed to sample from the mechanism. This makes the learned-adversary threat realistic whenever a system exposes perturbed outputs via a service.
>
> Furthermore, in the revised version, we add a learning-curve experiment for the learned adversary in **Appendix F.4**, where we subsample the attacker’s training data to different sizes and plot both (i) attack accuracy (measured as cosine similarity between reconstructed and ground-truth embeddings) and (ii) the mPL violation ratio (see new **Fig. 9 in Appendix F.4**). The curves show that the accuracy of attack saturates quickly: using only about 60\% of the supervised training pairs already attains nearly the same cosine accuracy and mPL violation ratio as training on the full dataset.
>
> We also observe that the estimated violation ratio *decreases slightly* as the training set grows. Our posterior-leakage metric is defined in terms of how much the posterior odds ratio $\frac{q_\theta(x_i \mid y)}{q_\theta(x_j \mid y)}$
> deviates from the prior odds $\frac{p(x_i)}{p(x_j)}$. With more supervised pairs, the two attack models become better calibrated, so their posterior ratios move closer to the true posterior odds and the estimated posterior leakage converges to its true value. With limited data, the learned scores are noisier and can be over-confident for some triples $(x_i, x_j, y)$, which can artificially inflate the deviation between posterior and prior odds and hence the apparent leakage. This provides a natural explanation for the slight downward trend in the violation curve as training data increase.
>
> **Weakness 3: Assumed knowledge of the embedding method.**
>
> **Response:** We intentionally adopt a strong attacker who knows the victim’s embedding model, following standard white-box practice in privacy and robustness work; this yields conservative, “best-case for the adversary” estimates of joint leakage. This assumption is realistic for many deployments where the encoder is public (e.g., off-the-shelf open models or advertised API backends), and even when it is not, an attacker can often train or reuse a surrogate encoder whose representations transfer reasonably well. Our attack framework itself does not require exact encoder knowledge: using a mismatched embedding model simply weakens the attacker, so our reported violation rates should be interpreted as upper bounds on what less-informed adversaries can achieve. In the following tables, we compare mPL under matched vs.\ mismatched embeddings across different threat models; the results show that with mismatched embeddings, the attacker achieves lower posterior leakage and therefore a smaller mPL violation ratio.
>
>
> ### RNN
>
> | metric                       | 0.30                | 0.40                | 0.50                |
> |------------------------------|---------------------|---------------------|---------------------|
> | match_posterior_leakage      | 0.1208±0.0141     | 0.1207±0.0063     | 0.1214±0.0113     |
> | mismatch_posterior_leakage   | 0.1035±0.0148     | 0.1075±0.0240     | 0.1007±0.0132     |
>
> ### LSTM
>
> | metric                       | 0.30                | 0.40                | 0.50                |
> |------------------------------|---------------------|---------------------|---------------------|
> | match_posterior_leakage      | 0.0956±0.0104     | 0.0934±0.0074     | 0.0951±0.0097     |
> | mismatch_posterior_leakage   | 0.0664±0.0190     | 0.0642±0.0144     | 0.0623±0.0231     |
>
> ### TRANSFORMER
>
> | metric                       | 0.30                | 0.40                | 0.50                |
> |------------------------------|---------------------|---------------------|---------------------|
> | match_posterior_leakage      | 0.1206±0.0075     | 0.1191±0.0059     | 0.1180±0.0086     |
> | mismatch_posterior_leakage   | 0.1183±0.0202     | 0.1183±0.0207     | 0.1182±0.0242     |

---

> ### Author Response · Authors · 2025-11-27
>
> Dear Reviewer,
>
> We sincerely appreciate the time and care you have devoted to reviewing our paper. As the discussion period draws to a close, we would like to kindly check whether there are any remaining concerns we may not have fully addressed. Your feedback has been extremely valuable in improving our work, and if you have any additional comments or suggestions, we would be very grateful for the opportunity to respond to them.

---

### Official Review · Reviewer_bgUX · 2025-10-29

**Soundness:** 3
**Presentation:** 3
**Contribution:** 3
**Rating:** 6
**Confidence:** 2

**Summary:**

The authors address the problem of metric differential privacy (mDP). Since mDP mechanisms do not account for joint observation, privacy leakage may occur when only individual records are considered. To address this, the authors formalize metric-normalized posterior leakage (mPL) and propose PBmPL as a framework to control it. However, since mPL cannot be computed directly, they introduce an attacker model to approximate it and adapt the noise level accordingly to prevent violations, thereby balancing privacy and utility.

**Strengths:**

* They formalized the concept of metric-normalized posterior leakage (mPL) and investigated its properties.
* They observe that existing metric differential privacy (mDP) mechanisms fail to adequately protect privacy under joint observation.
* They propose a method to control and reduce metric-normalized posterior leakage (mPL) under joint observation.

**Weaknesses:**

* The violation rate does not decrease significantly compared to the baseline when using the proposed method.
* The paper estimates mPL by training an adversarial model and averaging the results over multiple sampled instances. However, this approach may introduce errors both from the adversarial model itself and from sampling variance, especially when the sample set is large. There is no analysis provided to quantify or bound these potential sources of error.
* The paper lacks a clear analysis or visualization of the trade-off between utility and violation rate. Presenting this relationship with a graph would make the results more intuitive and convincing.
* The proposed method appears to require more time for sampling and training compared to the baseline. A comparison of computational cost or runtime would therefore be necessary.

**Questions:**

* I am not very familiar with this area, but I wonder whether there are other approaches that consider individuals separately. If such methods exist, a comparison with them would be necessary.

---

> ### Author Response · Authors · 2025-11-21
> **Response to Reviewer bgUX**
>
> Thank you for providing constructive comments/suggestions. Below, we provide the response to the questions and comments.
>
> **Question 1: Other approaches that consider individuals separately**
>
> **Response:** Thank you for this valuable question. We agree that when records can be reliably grouped by individual users and the mechanism is designed with explicit per-user accounting, a per-user privacy budget is a natural way to mitigate composition across correlated records.
>
> However, such user grouping is not always known or trustworthy. In many text/embedding applications, the system lacks reliable user identifiers, and deciding whether a set of words or snippets refers to the same person or secret is itself part of the adversary’s inference task (e.g., linking posts across accounts or mentions across documents). In our case study, the public datasets do not contain user IDs, so constructing a per-user baseline would require additional grouping assumptions that are orthogonal to our threat model. A per-user budget presupposes that this partition is known and enforced by the mechanism, whereas our framework is deliberately user-agnostic and evaluates joint leakage over arbitrary correlated secrets. We will clarify this point and discuss per-user accounting as a complementary mitigation in settings where reliable user identifiers are available.
>
> In the revised paper, we have added a brief discussion **at the end of Section 2** and a more detailed treatment in **Appendix C.2**.
>
> **Weakness 1: The violation rate does not decrease significantly**
>
> **Response:** Thank you for this valuable comment. We would like to clarify that EM is already very close to the ideal
> $exp(-\epsilon d)$ distribution, so its baseline violation rate is near the “noise floor’’ induced by finite samples and implicit correlations, leaving limited room for AmPL to further reduce violations without noticeably hurting utility. AmPL therefore makes only small corrections when the starting mechanism is already near mPL. In future work, we plan to explore richer mechanism families and joint training of the mechanism and auditor (rather than post-hoc reweighting of EM) to further tighten violation rates while still maintaining competitive utility.
>
> **Weakness 2: Approach may introduce errors**
>
> **Response:** We agree that, since our audit uses a learned surrogate for the posterior, it is important to make explicit how approximation error impacts the reported mPL. In the revision, we add **Proposition 5**, which provides an explicit bound on the gap between the true mPL and the surrogate mPL; the full proof is given in **Appendix D.5**.
>
> **Proof sketch of Proposition 5.** Training the adversary by minimizing conditional cross-entropy yields the standard decomposition: the population loss equals the entropy of the true posterior plus the expected KL divergence. Thus, minimizing cross-entropy is equivalent to minimizing this expected KL. We then show that the mPL expression is a Lipschitz function of the posterior probabilities (it is essentially a clipped log-odds over the probability simplex), so small posterior error---measured in total variation or via KL---translates into proportionally small error in mPL. Combining these observations with standard generalization bounds for deep networks implies that the expected difference between the true mPL (from $p$) and the surrogate mPL (from $q_\theta$) decreases polynomially in the number of training pairs.
>
> **Weakness 3: Tradeoff between utility and violation rate.**
>
> **Response:** We agree that explicitly visualizing the privacy--utility trade-off makes the results more intuitive. In the revision, we add **Figure 8** in **Appendix F.3**, which plots the empirical mPL violation ratio against utility loss for AmPL.  The figure shows that configurations with lower violation ratios tend to incur higher utility loss, while configurations that inject less noise achieve lower utility loss at the cost of higher violation ratios. This provides a direct and quantitative visualization of the expected privacy--utility trade-off and shows how AmPL can be tuned along this curve by adjusting $(\alpha_1,\alpha_2)$.
>
> **Weakness 4: Comparison of computational cost**
>
> **Response:** We agree that computational cost is relevant and will add a runtime comparison. In our prototype, AmPL requires about 565.6 seconds in total (data loading, preprocessing, and optimization), corresponding to 3.52±0.16 seconds per pipeline round over 153 rounds, whereas EM itself runs in under 1 second once embeddings are available. Importantly, this overhead is entirely offline: AmPL is calibrated once per dataset/mechanism/$\epsilon$, and online sampling only involves drawing from a precomputed categorical distribution, with essentially the same per-query cost as EM. In future work, we plan to reduce offline cost via more efficient hyperparameter search and reusing trained adversaries across nearby settings.

---

> ### Author Response · Authors · 2025-11-27
>
> Dear Reviewer,
>
> We sincerely appreciate the time and care you have devoted to reviewing our paper. As the discussion period draws to a close, we would like to kindly check whether there are any remaining concerns we may not have fully addressed. Your feedback has been extremely valuable in improving our work, and if you have any additional comments or suggestions, we would be very grateful for the opportunity to respond to them.

---

### Official Review · Reviewer_UW4E · 2025-10-30

**Soundness:** 3
**Presentation:** 3
**Contribution:** 2
**Rating:** 2
**Confidence:** 4

**Summary:**

This paper introduces metric-normalized posterior leakage, mPL, which is a distance-calibrated measure of how much an output shifts posterior odds between candidate secrets. mPL addreses leakage that arises when perturbed outputs are jointly consumed by the adversary, which is a setting where per-reccord metric mDP can be misleading.


In addition, this paper propose PBmPL, which bounds the frequency with which mPL may exceed a budget and supports estimation through sampling with a concentration guarantee. Moreover, they operationalize a trust-and-verify pipeline (AmPL) that (i) applies level-wise perturbations, (ii) trains a learned attacker to approximate posteriors and audit mPL, (iii) adapts mechanism strength from audit feedback, and (iv) optionally performs Bayesian remapping as pure post-processing.

In a word-embedding case study, they show that standard mDP mechanisms can still exhibit notable mPL violations under neural attackers, while AmPL substantially lowers the violation rate with comparable utility.

**Strengths:**

The proposed metric-normalized posterior leakage (mPL) is a novel privacy notion. mPL establishes basic properties, such as post-processing invariance, and proves that for single or independent releases, a uniform mPL bound is equivalent to mDP.


The adaptive mPL (AmPL) provides a concrete recipe, which is a creative combination of known pieces that makes an otherwise intractable problem operational for practitioners.

By centering joint observation of correlated items, the work hits a practically important gap that affects text embedding pipelines and other modern ML settings.

The case study with word embeddings demonstrates that mechanisms tuned for per-record mDP can still suffer non-trivial mPL violations under learned joint attackers, while AmPL materially reduces the violation frequency at comparable utility.

**Weaknesses:**

My main concern is as follows.

The paper's "certificate" relies on a **learned posterior surrogate** and a **sampling-based audit** (their PBmPL). It lacks (i) a **surrogate-to-truth transfer bound** (uniform over outputs) on likelihood ratios/privacy loss, (ii) **attacker model generalization control** (i.e., validity under worst-of-many attackers with proper multiple-comparison corection and hold-out evaluation), and (iii) any **composition** accounting across multiple tokens in a sequence of multiple releases. As a result, it seems that the claimed "certifiable protection under joint consumption" does not extend beyond the audited samples and the specific attacker or beyond a single run, even with infinite data/samples.


The post-processing invariance does not rescue the missing bounds. Remapping preserves whatever bound you already have. It does not creat a sequence-level or worst-case gurantee on its own.



# W-1: (i)

In this paper, they train a model to approximate posteriors (or distances mapped through a temperature-scaled softmax) and compute mPL from that surrogate.

Without a **uniform approximation bound** between the **true likelihodd ratio** (or the privacy-loss random variable) and its **surrogate**, the audit can under-estimate leakage even with infinite samples. Bayesian posterior-odds guarantees follow directly from likehood-ratio bounds. If those are only approximated, we must quantify the approximation error.


# W-2: (ii)

In this paper, they audit privacy leakage against one or a few learned attackers, with hyperparameter tuning, then report low violation rates. This might be a problem: Security claims should hold against a **class** of attack models, not just the one trained. Hyper/architecture search introduces multiple comparions and adaptivity to the audit set, which can hide violations (overfitting to the evaluation).



# W-3: (iii)

In this paper, they report per-mechanism or per token violation rates and improvements after the adaptive audit loop. Then discuss joint consumption.

The theoretical results are for single release (and independent releases), which are mathematically correct. The composition is not needed to validate those theoretical results.

However, the paper's central promise is privacy under joint consumption/repeated use. To claim that, we n**eed a composition argument** (or an explicit reduction to a composed privacy-loss bound). Without it, the paper's main claim is **under-justifed**. So, composition is not some add-on to paper's contribution; instead, it is required to elevate the scopr from single/independent to the realistic seting the paper cares about, especially under dependence.

**Questions:**

Please refer to the comments under Weaknesses.

---

> ### Author Response · Authors · 2025-11-21
> **Response to Reviewer UW4E**
>
> Thank you for providing constructive comments/suggestions. Below, we provide the response to the questions and comments.
>
> **Weakness 1: W-1: (i).**
>
> **Response:** We agree that, since our audit relies on a learned surrogate for the posterior, it is important to reason explicitly about how approximation error can affect the reported mPL. In the revised version, we add **Proposition 5** to estimate the approximation error bound between the true mPL and its surrogate, where the detailed proof is given in **Appendix D.5**.
>
> *Proof sketch of Proposition 5:* We note that training the adversary by conditional cross-entropy minimization yields a standard decomposition: the population loss equals a constant term (the entropy of the true posterior) plus the expected KL divergence between the true posterior $p(·|y)$ and the learned surrogate $q_\theta(·|y)$. Thus, minimizing cross-entropy is exactly minimizing the expected $KL(p || q_\theta)$. We then show that the mPL expression we compute is a Lipschitz function of the posterior probabilities (it is essentially a log-odds term over a probability simplex with mild clipping), so small error in the posterior, measured in total variation or via KL, induces proportionally small error in the resulting mPL values. Combining these facts gives an explicit bound: under standard generalization assumptions for deep networks, the expected mPL error between the true mPL (computed from $p$) and the surrogate mPL (computed from $q_\theta$) decreases polynomially with the number of training pairs.
>
> **Weakness 2: W-2: (ii).**
>
> **Response:** We agree that security claims should be made with respect to a class of attack models, not a single trained instance. Our goal is precisely to provide a flexible framework that can adapt to different threat models by plugging in different adversary classes, and in our experiments we instantiate three such classes: RNN-, LSTM-, and Transformer-based attackers. Formally, mPL and PBmPL are defined for an arbitrary adversary, and for any fixed adversary class our sampling procedure enjoys a standard concentration guarantee: the empirical violation rate converges to the true PBmPL violation probability as the number of samples grows. In practice, we use high-capacity neural estimators as strong but approximate proxies for Bayes-optimal attackers.
>
> We have clarified this perspective in the revised paper by adding a “**Scope of the audit**” paragraph right after **Proposition 5**, explicitly noting that our guarantees are conditioned on the chosen adversary class and positioning our framework as an adaptive auditing tool rather than a universal upper-bound guarantee over all possible attackers.
>
> **Weakness 3: W-3: (iii).**
>
> **Response:** We thank the reviewer for raising this important ***composition issue***. Our work is designed to capture correlation across multiple secrets within a single joint release: the attacker observes a batch of correlated tokens/records at once, and mPL is evaluated on this joint posterior. We do not model temporal composition over multiple releases of the same target, and a general composition theorem in this setting is, to our knowledge, still open. We fully agree that when the mechanism designer does know that multiple PIIs/PoIIs belong to the same person (e.g., via a reliable user identifier), one can define that person as the “target’’ and apply standard DP-style composition theorems over repeated releases for that target. In our text/embedding setting, by contrast, such targets are latent and secrets about different targets can be correlated (e.g., “Alice’s mother is a doctor” leaks about both Alice and her mother).
>
> We would like to clarify that user grouping is not always known in our setting. In many text/embedding applications, there is no reliable user or entity identifier, and determining which sensitive words refer to the same underlying secret is itself part of the adversarial inference problem. In our case study, the public datasets we use do not provide such identifiers, so any composed privacy-loss bound would necessarily rely on additional modeling assumptions about how records are grouped into targets. While these assumptions may be reasonable in some systems, they are not directly supported by our data and may not be robust to the very attacks we aim to model.
>
> Our contribution toward “joint use” is therefore operational rather than a new analytic composition bound: our theorems justify mPL as a single-release privacy-loss metric for a fixed attacker, and our adaptive audit loop measures and reduces joint violations on that attacker’s posterior over all correlated secrets in the release. In the revised paper, we have incorporated this clarification into the second paragraph of “**Scope of the audit**”. We have also added **Appendix C.2** to discuss per-user accounting as an alternative setting where classical composition results can be more directly applied.

---

> ### Author Response · Authors · 2025-11-27
>
> Dear Reviewer,
>
> We sincerely appreciate the time and care you have devoted to reviewing our paper. As the discussion period draws to a close, we would like to kindly check whether there are any remaining concerns we may not have fully addressed. Your feedback has been extremely valuable in improving our work, and if you have any additional comments or suggestions, we would be very grateful for the opportunity to respond to them.

---

### Official Review · Reviewer_xef9 · 2025-10-31

**Soundness:** 3
**Presentation:** 2
**Contribution:** 2
**Rating:** 4
**Confidence:** 3

**Summary:**

This paper challenges the assumption in metric differential privacy (mDP) that secrets are independent across records. The authors introduce metric-normalized posterior leakage (mPL), which quantifies how an adversary’s posterior belief shifts when observing multiple correlated releases. They prove that mPL is equivalent to mDP when secrets are released individually or independently distributed, but it can expose violations under joint observation.

To enforce mPL in practice, the authors propose Adaptive mPL: a trust-and-verify pipeline where a learned neural adversary estimates leakage and informs parameter adaptation to achieve probabilistically bounded mPL. Experiments on text embedding protection show that per-record mDP can fail under joint observation, while AmPL reduces such leakage with minimal utility loss.

**Strengths:**

* The work identifies a real gap in mDP deployment assumptions: joint inference over correlated records is typical in modern systems.
* mPL is clearly defined and theoretically grounded, recovering mDP under independent settings.
* The experiments provide useful evidence that per-record privacy guarantees do not prevent aggregate leakage when records are correlated.

**Weaknesses:**

* mPL application depends entirely on a learned adversary. While a strong adversary is assumed (with access to the noise mechanism and auxiliary data distribution), it does not necessarily represent an upper bound - a more efficient or capable adversary could still be possible. The claimed guarantees are therefore empirical rather than theoretical.
* Experiments focus solely on correlated records belonging to a single user. While I understand the general threat from correlated data, for the scenarios presented in the experimental section a per-user privacy budget could be sufficient to eliminate violations. I believe it should be at least incorporated as a baseline.
* The privacy budget range (0.3-0.5) is narrow, and results appear qualitatively similar across values. More extreme privacy regions would make trends clearer.
* Paper presenetation:
	* Initial empirical results (line 258) appear too early and are hard to follow at that stage of the paper.
	* Some figure labels are too small to read comfortably (e.g. Figure 2 and Figure 4).
	* Units for epsilon (km^-1) should be introduced earlier and explained clearly.
	* Sensitivity tiers are introduced without explanation.
	* The paper would benefit from a related works section

**Questions:**

What is the intuition for higher epsilon corresponding to lower violation rates? (Table 1)

---

> ### Author Response · Authors · 2025-11-21
> **Response to Reviewer xef9 (part 2)**
>
> **Weakness 3: The privacy budget range (0.3-0.5) is narrow.**
>
> **Response:** We agree that it is useful to understand the performance of our approach over a broader privacy budget range. Our experiments focus on the high-privacy regime because this is where joint-leakage violations are most concerning and where our method provides the most benefit. In fact, a key empirical observation is that once $\epsilon \geq 0.6$, the empirical violation ratios of all mechanisms already become very small, so the relative improvement from AmPL is negligible. In contrast, for $\epsilon \in [0.3, 0.5]$, existing mechanisms still exhibit nontrivial violations while AmPL substantially reduces them, which is precisely the regime of interest for strong privacy guarantees. We will clarify this rationale in the paper and add extended results over a wider $\epsilon$ range in the appendix to make these trends explicit. The following three tables report the posterior-leakage violation ratios of different methods for the AG News dataset when $\epsilon = 0.6, 0.7, 0.8, 0.9, 1.0$.
>
>
> ### RNN (%)
>
> | Method   | ε (km⁻¹) | 0.60        | 0.70        | 0.80        | 0.90        | 1.00        |
> |---------|----------|-------------|-------------|-------------|-------------|-------------|
> | EM (mDP) |          | 0.52±0.26 | 0.25±0.20 | 0.14±0.14 | 0.06±0.04 | 0.03±0.03 |
> | AmPL-U  |          | 0.51±0.32 | 0.24±0.22 | 0.15±0.15 | 0.06±0.04 | 0.04±0.03 |
> | AmPL-1  |          | 0.41±0.21 | 0.11±0.06 | 0.02±0.02 | 0.00±0.00 | 0.00±0.02 |
> | AmPL    |          | 0.35±0.21 | 0.14±0.11 | 0.08±0.06 | 0.04±0.02 | 0.02±0.04 |
>
> ---
>
> ### LSTM (%)
>
> | Method   | ε (km⁻¹) | 0.60        | 0.70        | 0.80        | 0.90        | 1.00        |
> |---------|----------|-------------|-------------|-------------|-------------|-------------|
> | EM (mDP) |          | 0.21±0.13 | 0.07±0.07 | 0.02±0.03 | 0.01±0.01 | 0.00±0.00 |
> | AmPL-U  |          | 0.21±0.13 | 0.07±0.08 | 0.02±0.03 | 0.01±0.01 | 0.00±0.00 |
> | AmPL-1  |          | 0.14±0.16 | 0.01±0.03 | 0.00±0.01 | 0.00±0.00 | 0.00±0.00 |
> | AmPL    |          | 0.14±0.06 | 0.05±0.06 | 0.02±0.03 | 0.01±0.01 | 0.00±0.00 |
>
> ---
>
> ### TRANSFORMER (%)
>
> | Method   | ε (km⁻¹) | 0.60        | 0.70        | 0.80        | 0.90        | 1.00        |
> |---------|----------|-------------|-------------|-------------|-------------|-------------|
> | EM (mDP) |          | 0.69±0.33 | 0.31±0.16 | 0.11±0.08 | 0.06±0.05 | 0.04±0.07 |
> | AmPL-U  |          | 0.68±0.32 | 0.32±0.19 | 0.12±0.08 | 0.06±0.06 | 0.04±0.07 |
> | AmPL-1  |          | 0.38±0.36 | 0.07±0.10 | 0.00±0.00 | 0.00±0.00 | 0.00±0.00 |
> | AmPL    |          | 0.43±0.11 | 0.18±0.07 | 0.06±0.04 | 0.03±0.03 | 0.02±0.02 |
>
>
>
> **Weakness 4: Paper presentation:**
>
> **Response:**
>
> (1) We have moved the description of initial empirical results to the first paragraph of **"Main Results" in Section 4**.
>
> (2) Labels in Figure 2 (current Figure 3) and Figure 4 have been enlarged.
>
> (3) Epsilon (km^-1) was a typographical error, and the unit (km^-1) has been removed in the revised paper.
>
> (4) We have added the introduction of sensitivity tiers (in **the second paragraph of Section 3**).
>
> (5) We have added the related work section in **Appendix G**.

---

> ### Author Response · Authors · 2025-11-21
> **Response to Reviewer xef9 (part 1)**
>
> Thank you for providing constructive comments/suggestions. Below, we provide the response to the questions and comments.
>
> **Question 1: Intuition for higher epsilon corresponding to lower violation rates**
>
> **Response:** The empirical violation ratio is computed by checking whether posterior odds exceed the bound $\exp(\epsilon d(s,s'))$. When $\epsilon$ is low, this bound is very tight, so even modest deviations introduced by implicit correlation across releases can cause a pair to be flagged as a violation. As $\epsilon$ increases, the allowable bound $\exp(\epsilon d(s,s'))$ grows faster than these correlation-induced deviations, so the same mechanism lies more comfortably inside the feasible region and fewer pairs are counted as violations. Empirically, we observe exactly this behavior: $\epsilon \geq 0.6$, all mechanisms already exhibit very low violation ratios, and the additional improvement from AmPL becomes negligible (Please find this observation in the **response to Weakness 3**).
>
> We have added this clarification to **the first paragraph of “Main Results” in Section 4**.
>
> **Weakness 1: mPL application depends entirely on a learned adversary.**
>
> **Response:** We agree that no fixed learned adversary can represent a universal upper bound over all conceivable attackers. This limitation is inherent when working with complex DNN–based adversaries, where computing the posterior (and thus exact mPL) is computationally infeasible. *Our goal is therefore not to provide a universal worst-case upper bound, but to design a flexible framework that can adapt to different threat models by plugging in different adversary classes.*
>
> Formally, mPL and PBmPL are defined for an arbitrary adversary, and for any fixed adversary class our sampling procedure enjoys a concentration guarantee: the empirical violation rate converges to the true PBmPL violation probability as the number of samples grows. In practice, we instantiate this with high-capacity neural estimators, which serve as strong but necessarily approximate proxies for Bayes-optimal attackers. Stronger or alternative adversaries can always be plugged into the same audit loop, potentially revealing additional violations.
>
> We have clarified this as "**scope of audit**" in the revised paper in **right after Proposition 5**.
>
> **Weakness 2: Experiments focus solely on correlated records belonging to a single user.**
>
> **Response:** We agree that when records can be cleanly grouped by user and the mechanism is designed with explicit per-user accounting, a per-user privacy budget is a natural mitigation for composition across correlated records.
>
> However, *user grouping is not always known or trusted.* In many text/embedding applications, the system does not have a reliable user identifier. Determining that a set of words or snippets describe the same person/secret is itself part of the adversarial inference problem (e.g., linking posts across accounts or mentions across documents). In our case study, the public datasets we use do not contain user identifiers, so constructing a per-user baseline would require additional grouping assumptions that are orthogonal to our threat model. A per-user budget presupposes that this grouping is known and enforced by the mechanism, whereas our framework is deliberately user-agnostic and targets joint leakage over arbitrary correlated secrets. We will clarify this point and discuss per-user accounting as a complementary mitigation when reliable user IDs are available.
>
> We have added a brief discussion at the end of **Section 2** in the revised paper and provided a more detailed treatment in **Appendix C.2**.

---

> > ### Comment · Reviewer_xef9 · 2025-11-26
> >
> > I thank the authors for their responses.
> > I am satisfied with the expanded results and clarifications, specifically "scope of audit" section in the revised paper. Authors correctly identify the limitation of their method, and only claim to provide bounds for a specific class of attackers. While the claim is quite narrow, and the method would not be applicable for the context requiring guarantees for all possible attackers and input distributions, I believe the paper makes an good contribution, and supports claims its making.
> >
> > I will therefore increase my score.

---

> > > ### Author Response · Authors · 2025-11-26
> > > **Response to Reviewer xef9**
> > >
> > > Thank you very much for taking the time to reconsider our submission and for carefully engaging with the revised “scope of audit” section and additional results. We sincerely appreciate your constructive feedback and your updated assessment of the paper.

---

### Author Response · Authors · 2025-12-04
**Summary of Rebuttal Outcomes and Paper Revisions**

Dear Area Chairs and Senior Chairs,

Frist, we would like to sincerely thank you and all the reviewers for the time and effort invested in evaluating our submission, and for the constructive feedback during the rebuttal and discussion phase.

After the rebuttal and revision, **Reviewer xef9** increased their score **from 4 to 6**, so the overall scores changed from **4, 2, 6, 4** to **6, 2, 6, 4**. Although the other reviewers had not updated their reviews before the discussion phase was frozen, we believe we have carefully addressed their main concerns in both the response and the revised manuscript. We have made every effort to accurately represent their remarks.

Below we briefly summarize our main responses and where they can be found in the revised paper:


-------

***1. Scope, threat model, and composition (Reviewers xef9, UW4E, bgUX, ynkJ)***: We clarified the intuition behind how violation rates depend on $\epsilon$ and moved the key empirical takeaway into the first paragraph of **“Main Results” in Section 4**. We added a dedicated **“Scope of the audit”** paragraph right after **Proposition 5**, explicitly stating that our guarantees are conditioned on a chosen adversary class and positioning our method as an adaptive auditing tool.

We also discuss per-user privacy budgets vs. adversarial aggregation of correlated records at **the end of Section 2**, with a more detailed treatment in **Appendix C.2**. We clarified the generality of the framework beyond text (and why this paper focuses on text embeddings) at the end of **the first paragraph of Section 4**, with an extended discussion in **Appendix C.3**.

***2. Approximation error of the learned adversary (Reviewers UW4E, bgUX)***: We introduced **Proposition 5 in Section 4**, which provides an explicit bound on the gap between the true mPL and the mPL computed using the learned surrogate posterior. **The full proof of Proposition 5 is given in Appendix D.5**.

The **“Scope of the audit” paragraph after Proposition 5** also clarifies that all empirical violation guarantees are attacker-class–specific and do not claim a universal bound over all possible attackers.

***3. Experimental coverage and practicality (Reviewers xef9, UW4E, bgUX, ynkJ)***:

- **Extended $\epsilon$ range on AG News**: In the response to **Reviewer xef9 (part 2)**, we added new tables that report mPL violation ratios on AG News for a wider $\epsilon$ range. These results show that all mechanisms’ violation ratios decrease as $\epsilon$ grows and become very small in the low-privacy regime, while AmPL consistently achieves the lowest violation rates.
- **Privacy–utility trade-off visualization**: We also added a new privacy–utility trade-off plot (new **Figure 8 in Appendix F.3**), which plots mPL violation ratio vs. utility loss for AmPL. This makes explicit how increasing noise reduces violations at the cost of utility, and how AmPL can be tuned along this curve.
- **New tabular dataset experiment**: To demonstrate generality beyond text, in the **response to Reviewr ynkJ**, we added an experiment on the Breast Cancer Wisconsin (Diagnostic) tabular dataset. The new tables compare EM (mDP) and AmPL under multiple attack models and show that AmPL’s posterior leakage is 58.1% lower than EM on average when applied to protect non-text data.
- **Learning-curve experiment for the attacker**: To address concerns about training-data requirements, we added a learning-curve experiment (new **Figure 9 in Appendix F.4**). We subsample the attacker’s training pairs to different fractions and plot both: (i) attack accuracy (measured via cosine similarity between reconstructed and true embeddings), and (ii) mPL violation ratio. The curves show that accuracy saturates quickly, with only about 60% of the training data achieving almost the same accuracy and violation ratio as using the full dataset, and that the estimated violation ratio decreases as the attacker becomes more accurate.
- **Runtime and computational cost**: In the **response to Reviewer bgUX**, we added a runtime analysis, reporting the total offline calibration time and per-round cost. For AG News, AmPL’s offline calibration (including data loading, preprocessing, and optimization) takes about 633 ± 26 seconds in total, corresponding to about 3.52 ± 0.16 seconds per pipeline round over 180 rounds. We emphasize that this cost is entirely offline: once calibrated for a given dataset/mechanism/epsilon, AmPL has essentially the same per-query online cost as EM, and we discuss how future work could further reduce this overhead (e.g., more efficient hyperparameter search and reusing trained attackers across nearby settings).

------

Some of the lower scores appear to stem from misunderstandings and an earlier, incomplete set of experiments and theoretical analyses. We have substantially strengthened both in the revision and respectfully ask for a re-evaluation that fully considers our rebuttal and updated manuscript.

---

### Meta-Review · Area_Chair_KKWK · 2026-01-04

**Summary:**

This paper addresses a gap in Metric Differential Privacy (mDP): the assumption that secrets are independent across records. In reality, correlated data releases can inadvertently leak information that traditional mDP fails to capture.

The authors introduce mPL (metric-normalized Posterior Leakage), a new privacy metric that quantifies the shift in an adversary’s posterior belief after observing correlated releases. The authors prove that mPL is mathematically equivalent to mDP when secrets are independent. mPL exposes privacy violations that mDP misses when data is viewed collectively.

**Reviewer Concerns:**

Reviewer xef9 was satisfied with the expanded results during the rebuttal and clarifications, specifically "scope of audit" section in the revised paper.

Reviewer UW4E identified the main concerns with the paper, namely: "The paper's "certificate" relies on a learned posterior surrogate and a sampling-based audit (their PBmPL). It lacks (i) a surrogate-to-truth transfer bound (uniform over outputs) on likelihood ratios/privacy loss, (ii) attacker model generalization control (i.e., validity under worst-of-many attackers with proper multiple-comparison correction and hold-out evaluation), and (iii) any composition accounting across multiple tokens in a sequence of multiple releases." The authors tried to address the issues and even added additional proofs. However, these additions require further verification, thus, the paper should be re-submitted with the improvements.

A similar concern to reviewer xef9 was also raised by reviewer bgUX. Again, the new additions should be thoroughly verified. Reviewer bgUX also pointed out to the issue with the substantial computation overhead compared to the EM baseline. The authors should reduce the offlie cost before the re-submission.

The last reviewer ynkJ also raised many points that authors addressed in detail. However, only the first reviewer responded and it is extremely hard to judge if the provided responses fully addressed all the comments.

Based on the evaluations from reviewers UW4E and xef9, the paper requires additional validation before it can be accepted. I recommend that the authors incorporate the reviewers' suggestions into an updated version and re-submit the work for a new round of review.

**Reviewer Scores:**

Reviewer xef9: Score increased probably from 4 to 6 / Confidence 3.

Reviewer UW4E: Score 2 / Confidence 4

Reviewer bgUX: Score 6 / Confidence 2

Reviewr ynkJ: Score 4 / Confidence 4

Reviewer jfMJ: not provided

---

### Decision · Program_Chairs · 2026-01-26

Reject